# Medium-Difficulty Samples Constitute Smoothed Decision Boundary for Knowledge Distillation on Pruned Datasets

**Yudong Chen**[1,2]    **Xuwei Xu**[1,2]    **Frank de Hoog**[2]    **Jiajun Liu**[2,1*]    **Sen Wang**[1*]
[1]The University of Queensland    [2]CSIRO Data61
{yudong.chen,xuwei.xu,sen.wang}@uq.edu.au
{jiajun.liu,frank.dehoog}@csiro.au

## Abstract

This paper tackles a new problem of dataset pruning for Knowledge Distillation (KD), from a fresh perspective of Decision Boundary (DB) preservation and drifts. Existing dataset pruning methods generally assume that the post-pruning DB formed by the selected samples can be well-captured by future networks that use those samples for training. Therefore, they tend to preserve hard samples since hard samples are closer to the DB and better characterize the nuances in the distribution of the entire dataset. However, in KD, the limited learning capacity from the student network leads to imperfect preservation of the teacher's feature distribution, resulting in the drift of DB in the student space. Specifically, hard samples worsen such drifts as they are difficult for the student to learn, creating a situation where the student's DB can drift deeper into other classes and make incorrect classifications. Motivated by these findings, our method selects medium-difficulty samples for KD-based dataset pruning. We show that these samples constitute a smoothed version of the teacher's DB and are easier for the student to learn, obtaining a general feature distribution preservation for a class of samples and reasonable DB between different classes for the student. In addition, to reduce the distributional shift due to dataset pruning, we leverage the class-wise distributional information of the teacher's outputs to reshape the logits of the preserved samples. Experiments show that the proposed static pruning method can even perform better than the state-of-the-art dynamic pruning method which needs access to the entire dataset. In addition, our method halves the training times of KD and improves the student's accuracy by 0.4% on ImageNet with a 50% keep ratio. When the ratio further increases to 70%, our method achieves higher accuracy over the vanilla KD while reducing the training times by 30%. Code is available at https://github.com/chenyd7/MDSLR.

## 1 Introduction

Deep Neural Networks (DNNs) (Krizhevsky et al., 2012; Szegedy et al., 2015; He et al., 2016; Liu et al., 2022) have dominated the computer vision field in the past decade as an effective feature extraction tool. One of the most important factors in the success of DNNs is the large number of parameters. For instance, the early GoogLeNet (Szegedy et al., 2015) utilizes approximately 6.6 million parameters to obtain 69.77% top-1 accuracy on ImageNet (Russakovsky et al., 2015) classification, and the recently proposed ConvNeXt (Liu et al., 2022) with 197.8 million parameters increases the accuracy to 84.41%. Although large networks are powerful, they also significantly increase computation costs for inference, which limits their deployments in real-world applications.

To mitigate this problem, Knowledge Distillation (KD) (Hinton et al., 2015) is proposed to enhance the performance of lightweight networks for efficient inference. The mechanism of KD is to utilize the knowledge hidden in the teachers (large networks) to supervise the training of students (lightweight networks). Inspired by the promising performance of KD, a series of extensions have

---

*Corresponding Author

been proposed by exploiting knowledge in different layers of the teacher network (Ji et al., 2021; Chen et al., 2022) or designing more effective distillation loss (Tian et al., 2020; Zhao et al., 2022). Although distillation-based training can greatly improve lightweight networks' performance, the training complexity is high due to the addition of the teacher and the usage of large-scale datasets.

There are several methods designed to reduce the training cost of KD. For instance, self-distillation proposes to replace the teacher's knowledge with the latent knowledge hidden in the student itself (Sun et al., 2019; Zhang et al., 2019; Liang et al., 2022; Yang et al., 2023). Although the training times are reduced by discarding the cumbersome teacher network, the improvement of the student's accuracy is limited due to the intrinsic performance gap between the student and the teacher. Another solution for KD acceleration is to reduce the frequency of forward propagation (Shen & Xing, 2022; Beyer et al., 2022). A typical way is to store the knowledge generated by the pre-trained teacher in advance. However, since the technique of data augmentation plays an important role in KD, the fixed knowledge tends to be less effective as shown in (Beyer et al., 2022). Recently, dynamic dataset pruning methods have shown great potential in speeding up training (Li et al., 2022; Truong et al., 2023; Qin et al., 2024). The main idea of dynamic dataset pruning is to skip the less informative samples for forward propagation during training based on the loss values from the previous epoch. By focusing more on learning those hard samples, the training cost can be reduced and the distillation performance can be maintained to a large extent (Li et al., 2022). However, dynamic data pruning methods still need to access the entire dataset. As the sizes of training datasets constantly increase in the field of computer vision, the I/O operations for large-scale datasets will stress resource-constrained devices (Dryden et al., 2021; Nguyen et al., 2022).

To reduce the training cost and the size of datasets simultaneously, this paper explores how to perform static dataset pruning (Welling, 2009; Toneva et al., 2019; Paul et al., 2021) given a pre-trained teacher network. Specifically, static pruning decides which samples should be preserved at the beginning of training and does not need to store the entire dataset. However, existing static pruning methods are less effective in KD scenarios as illustrated in the recent research (Ben-Baruch et al., 2024) and Table 5. This is because these methods are not tailored for distillation and neglect the capacity gap between the student and the teacher. In general, static pruning methods tend to preserve hard samples for training as they are more informative and closer to the decision boundary. However, we observe the drift of decision boundary in distillation since the student is unable to fully mimic the teacher's feature distribution of hard samples, leading to the overlap of features of different classes. Based on this observation, we propose preserving medium-difficulty samples for distillation as they form a smoothed decision boundary of the original distribution and are easier to learn. In addition, to mitigate the distributional shift due to the lack of a large portion of samples, we record the average distribution information of the teacher's predictions for subsequent distillation. By reshaping the logits of the preserved samples with the recorded information, the student's performance can be further improved. In summary, the contributions of this paper include:

- We investigate the effects of using different types of samples for distillation from the perspective of decision boundary. To avoid the drift of decision boundary, we propose preserving the medium-difficulty samples for training.

- To mitigate the distributional shift due to dataset pruning, we use the average logit distribution information of the teacher to reshape the logits of preserved samples and modify the distillation loss for the training on a pruned dataset.

- Experimental results demonstrate that our method can even perform better than the state-of-the-art dynamic pruning method. In addition, when using 70% of samples, our method achieves higher accuracy over the vanilla KD while reducing the training times by 30%.

## 2 RELATED WORK

**Efficient knowledge distillation.** The original KD and its extensions generally require a pre-trained teacher network to generate soft labels (Hinton et al., 2015; Zhao et al., 2022) or intermediate features (Ji et al., 2021; Chen et al., 2022) to guide the student's training, which is computationally intensive due to the forward propagation of the teacher. In contrast, self-distillation aims to obtain knowledge without using a large teacher network. For instance, some methods add multiple classifiers to the student network to generate soft labels from the shallow features (Sun et al., 2019; Zhang et al., 2019). Since the additional branches inevitably introduce more parameters, the computation

costs are still high. Recent self-distillation methods have been proposed to discover the knowledge hidden in the student itself. Zipf's LS (Liang et al., 2022) reuses the classifier of the student to perform dense prediction on the feature maps before the pooling layer and obtains soft labels by ranking the frequency of the predicted class in the dense prediction. Universal Self-Knowledge Distillation (USKD) (Yang et al., 2023) further divides the distillation loss into non-target and target objectives based on Zipf's LS. Although Zipf's LS and USKD significantly reduce the training complexity of self-distillation, the distilled students are less discriminative than those using the knowledge from the large teacher. Since the teacher's knowledge is indispensable, another line of efficient KD is to reduce the times of forward propagation. An intuitive idea is to store the knowledge (e.g., soft labels) of different samples generated by the teacher before distillation as the weights of the teacher are fixed during training. However, this not only increases the costs for storing the fixed knowledge but also degrades the effectiveness of KD due to the lack of data augmentation and the inconsistency between the inputs of the student and the teacher at different epochs (Beyer et al., 2022). Therefore, this paper focuses on pruning the less important training samples for efficient distillation.

**Dataset pruning.** Most recently, dynamic dataset pruning methods have shown promising performance by selectively feeding the samples to the networks for optimization based on the loss from the previous epoch (Truong et al., 2023; Qin et al., 2024). The rationale behind dynamic pruning methods is that samples with larger losses are more informative for the student to learn. By eliminating the less important samples for forward propagation, the computation costs can be greatly reduced and the overall performance is surprisingly competitive (Qin et al., 2024). A drawback of dynamic pruning methods is that they need to access the entire dataset. As the sizes of datasets constantly increase in the trend of Scaling Laws (Alabdulmohsin et al., 2022), the I/O overhead of accessing data is likely to be a bottleneck in training a model (Dryden et al., 2021; Nguyen et al., 2022). Therefore, it is desirable to develop static pruning methods for efficient training. The difference between static and dynamic pruning is that the static methods prune samples in advance and only access the pruned dataset during training. Similarly, existing static pruning methods are prone to preserve hard samples. For example, Forgetting (Toneva et al., 2019) selects samples that are harder to memorize and Error L2-Norm (EL2N) (Paul et al., 2021) preserves samples that have larger discrepancies with the corresponding ground-truth labels. Since these static methods do not consider the capacity gap between the student and the teacher, their effectiveness in KD is degraded.

## 3 METHODOLOGY

In this paper, we aim to provide insights for practitioners that directly use the pre-trained models (e.g., downloading model weights from the repository of Pytorch[1]) to distil a student and try to prune the dataset using the prior knowledge from the pre-trained teacher without introducing too much computation overhead. The study on static dataset pruning for KD is limited. For instance, Ye et al. (2024) conducts a few-shot classification experiment based on the KD framework. However, the focus of (Ye et al., 2024) is to train an appropriate teacher for the student's distillation, which is orthogonal to the goal in this paper. The recent research (Ben-Baruch et al., 2024) proposes to adaptively adjust the hyper-parameters of distillation and classification losses according to the size of the pruned dataset. However, Ben-Baruch et al. (2024) suggests randomly pruning samples in KD scenarios. On the contrary, we propose to utilize medium-difficulty samples for distillation and achieve better performance. Our pruning strategy is similar to Moderate Dataset Selection (MoDS) (Xia et al., 2023), which aims at improving the robustness and generalization of dataset pruning by preserving the samples that are closer to the median distances to the corresponding class centres. Different from MoDS, we investigate the effects of using medium samples from the perspective of distillation-based training. In addition, we design a logit reshaping method for the improvement of distillation on pruned data as illustrated in the following subsections.

### 3.1 PRELIMINARIES

The vanilla KD uses the teacher's logits to guide the student's training. In this paper, the teacher's logit and the student's logit of a certain sample are denoted by $p \in \mathbb{R}^{c \times 1}$ and $z \in \mathbb{R}^{c \times 1}$, respectively, where $c$ is the number of classes. After obtaining the logits, the distillation loss used to train the

---

[1]https://pytorch.org/vision/stable/models.html

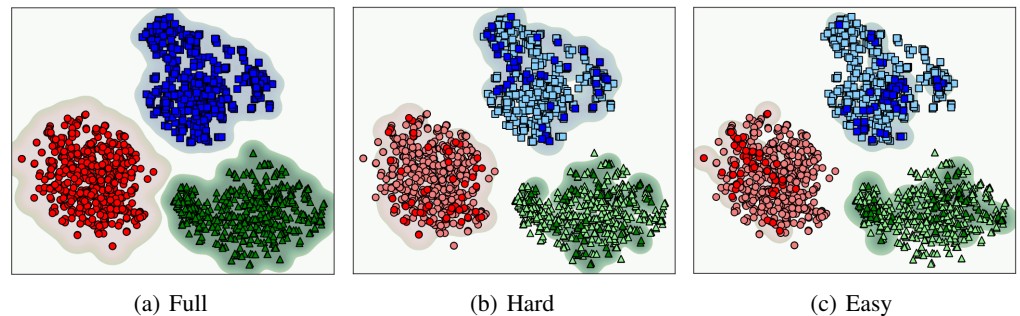

|  |  |  |
|---|---|---|
| (a) Full | (b) Hard | (c) Easy |

Figure 1: Decision boundaries of (a) full samples, (b) hard samples, and (c) easy samples selected by the pre-trained network ResNet50 on CIFAR-100. Different shapes denote samples of different classes and the selected samples are visualized with darker colours. Note that hard samples better preserve the original decision boundary, covering the general distribution of the entire dataset.

student is as follows:

$$\mathcal{L}_{KL} = -\sum_{i=1}^{c} \frac{\exp(p_i/\mu)}{\sum_j \exp(p_j/\mu)} \log \frac{\exp(z_i/\mu)}{\sum_j \exp(z_j/\mu)}, \quad (1)$$

where $p_i$ and $z_i$ are the $i$-th element of $p$ and $z$, respectively, and $\mu$ is the temperature coefficient. By combining the distillation loss with the cross-entropy loss $L_{CE}$, the total loss for the student's optimization is as follows:

$$\mathcal{L}_{KD} = \alpha\mathcal{L}_{CE} + (1-\alpha)\mathcal{L}_{KL}, \quad (2)$$

where $\alpha$ is a hyper-parameter used to trade off the two losses. As shown in Equation 2, the distillation-based training requires a pre-trained teacher network to generate logits for supervision, which will significantly increase computation costs. To mitigate this problem, a feasible way is to reduce the number of training samples. In KD scenarios, it is common to use the pre-trained teacher networks downloaded from different open-source repositories for distillation. Therefore, this paper aims to directly prune the dataset according to the predictions of the pre-trained teacher without re-training (Toneva et al., 2019) or introducing an additional proxy network (Coleman et al., 2020) so that the computation overhead for dataset pruning can be reduced. Existing dataset pruning methods score the difficulties of training samples by using different measurements (Paul et al., 2021; Qin et al., 2024). For example, EL2N obtains the score of a training sample by computing the L2 distance between the network's predictions and the one-hot label vectors. Similarly, this paper scores the samples by measuring the discrepancy between the network's predictions and the one-hot label vectors with the cross-entropy loss since we conduct experiments on the image classification task, which generally adopts the cross-entropy loss for optimization. Given a pre-trained teacher network, the difficulties of training samples are defined as follows:

$$\mathcal{M}_{DIF} = -\sum_{i=1}^{c} y_i \log \frac{\exp(p_i)}{\sum_j \exp(p_j)}, \quad (3)$$

where $y \in \mathbb{R}^{c\times 1}$ is a one-hot vector constructed from the ground-truth label. We rank the difficulty of training samples based on the values of $\mathcal{M}_{DIF}$ and obtain an index vector $d = \{d_1, d_2, ..., d_i, ..., d_n\} \in \mathbb{R}^{n\times 1}$, where $n$ is the total number of training samples and $d_i$ is the index of the training sample with the $i$-th largest loss. Samples with larger losses, i.e., $d^H = \{d_1, d_2, ..., d_i, ..., d_{rn}\} \in \mathbb{R}^{rn\times 1}$ are regarded as hard samples and samples with smaller losses, i.e., $d^E = \{d_{n-rn+1}, d_{n-rn+2}, ..., d_{n-rn+i}, ..., d_n\} \in \mathbb{R}^{rn\times 1}$ are considered as easy samples, where $0 < r < 1$ is the ratio of the preserved samples and we assume $rn$ is an integer for simplicity.

## 3.2 SMOOTHED DECISION BOUNDARY FROM MEDIUM-DIFFICULTY SAMPLES

Most of the existing dataset pruning methods tend to preserve hard samples (Toneva et al., 2019; Paul et al., 2021) for training as hard samples are informative while a few methods find the easy

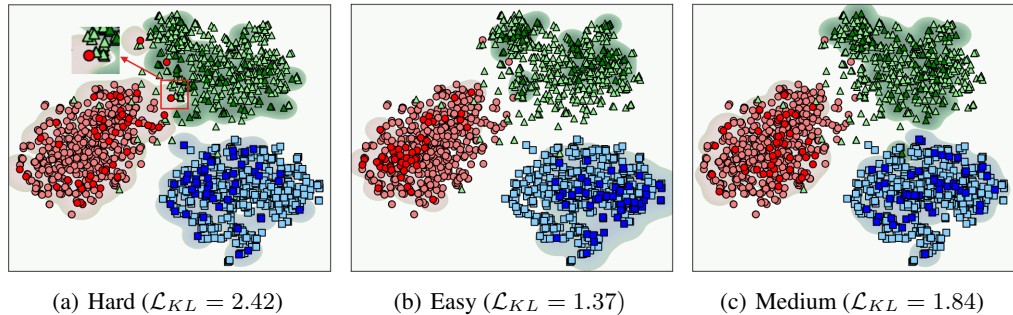

(a) Hard ($\mathcal{L}_{KL} = 2.42$)  (b) Easy ($\mathcal{L}_{KL} = 1.37$)  (c) Medium ($\mathcal{L}_{KL} = 1.84$)

Figure 2: Decision boundaries of (a) hard samples, (b) easy samples, and (c) medium samples selected by the teacher ResNet50 on the feature space of the distilled student MobileNetV2 on CIFAR-100. It is shown that the decision boundary of hard samples drifts into other classes and the decision boundary of easy samples is too small to cover the general distribution of the dataset. Medium-difficulty samples obtain a balance between these situations.

samples are beneficial (Welling, 2009). To compare the difference between hard and easy samples selected by the teacher network, we visualize their decision boundaries as shown in Figure 1. In these visualizations, we first map the deep features generated by the pre-trained ResNet50 (He et al., 2016) into a two-dimensional space by using t-SNE (Van der Maaten & Hinton, 2008) and then train a classifier with the selected samples to generate the decision boundary. Figure 1 demonstrates that the decision boundary obtained by hard samples is closer to that generated by the original full data. Since the feature distribution of easy samples is relatively concentrated, its decision boundary retreats too much from the original, which may affect the network's generalization ability. Figure 1 suggests that preserving hard samples is beneficial to dataset pruning. However, we find that the performance of existing static pruning methods is far from satisfactory in distillation-based training. We hypothesize this is because the student network is unable to learn the exact feature distribution of the teacher due to the large capacity gap, leading to the deviation of the decision boundary.

To verify this hypothesis, we first compare the training accuracy of the distilled student MobileNetV2 (Sandler et al., 2018) and the pre-trained teacher ResNet50. In Figure 3, we select three subsets from the training set based on the teacher's predictions, including 30% of hard samples, 30% of medium samples, and 30% of easy samples. It is shown that there is a large gap between the student and the teacher regarding the training accuracies of hard samples. This is because hard samples are close to the decision boundary of the corresponding class and tend to be misclassified. Since the student is unable to mimic the teacher exactly, the hard samples selected by the teacher may not be suitable for the student's training. As shown in Figure 2(a), some of the hard samples belonging to the "circle" class are indistinguishable from the "triangle" class. As a result, the decision boundary formed by the hard samples of the "circle" class involves several samples belonging to the "triangle" class, leading to incorrect classification. On the other hand, although easy samples are easier to learn according to the lower distillation loss, the range of the decision boundary of easy samples is small as shown in Fig-

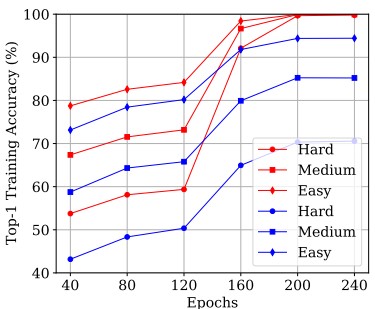

Figure 3: Top-1 training accuracy of different types of samples on CIFAR-100. The red lines represent the results obtained by the pre-trained ResNet50 and the blue lines represent the results obtained by the distilled MobileNetV2.

ure 2(b). Therefore, a large portion of data points are not included, which will affect the network's generalization ability. The above analysis indicates that neither hard nor easy samples are suitable for distillation to obtain accurate and generalizable decision boundaries for different classes. Motivated by these observations, we propose preserving the medium-difficulty samples, i.e., $d^M = \{d_{\lceil \frac{n-rn}{2} \rceil}, ..., d_{\lceil \frac{n}{2} \rceil}, ..., d_{\lceil \frac{n+rn}{2} \rceil - 1}\} \in \mathbb{R}^{rn \times 1}$ for distillation on a pruned dataset and em-

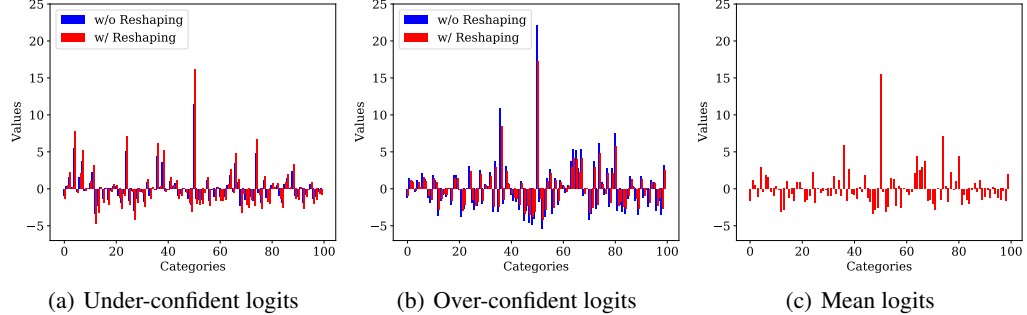

(a) Under-confident logits      (b) Over-confident logits      (c) Mean logits

Figure 4: Reshaping logits of the preserved samples using the class-wise distribution information of the entire dataset. Given (a) an under-confident logit vector or (b) an over-confident logit vector of the preserved sample, we reshape it to better fit the distribution of (c) the mean logit vector of the corresponding class on the full dataset.

pirically find that such dataset pruning strategy work well in different teacher-student combinations. Similarly, we visualize the decision boundary learned by the medium-difficulty samples in Figure 2(c). It is shown that the medium-difficulty samples constitute a smoothed decision boundary and are easier to learn than hard samples. In addition, the medium-difficulty samples obtain a trade-off between underfitting and overfitting on the training set compared to the hard and easy samples as illustrated in Figure 3. Therefore, the student's generalizability on the testing set is better by using medium-difficulty samples for distillation as shown in Section 4.1.

### 3.3   MITIGATING BIAS OF THE PRUNED DATA

Although the distillation on a pruned dataset can be improved by using medium-difficulty samples, it is still challenging to achieve lossless performance due to the information loss of a large portion of pruned samples. To mitigate this problem, we record the global distribution information of the teacher's predictions to reshape the logits of the selected samples. Specifically, we obtain the class-wise logit information of the $i$-th class as follows:

$$k_i = \frac{\sum_{j=1}^{m_i} \mathcal{M}_{STD}(p^{i,j})}{m_i} \tag{4}$$

where $m_i$ is the number of samples of the $i$-th class, $\mathcal{M}_{STD}(\cdot)$ is used to compute the standard deviation of a logit vector and $p^{i,j}$ is the logit vector of the $j$-th samples in $i$-th class. Since the recorded information $k$ is a $c$-dimensional vector, it barely increases the storage overhead. After obtaining the average logit information, we use it to reshape the logits of the preserved samples and rewrite the distillation loss as follows:

$$\tilde{\mathcal{L}}_{KL} = -\sum_{i=1}^{c} \frac{\exp(\tilde{p}_i/\mu)}{\sum_j \exp(\tilde{p}/\mu)} \log \frac{\exp(z_i/\mu)}{\sum_j \exp(z_j/\mu)}, \tag{5}$$

where $\tilde{p} = pk_p/\mathcal{M}_{STD}(p)$ and $k_p$ is the average logit information of the class that $p$ belongs to. The idea of KD is to use the soft labels generated by the teacher to characterize the latent between-class information of the dataset. Since the pruned dataset only preserves a portion of samples, the teacher may be biased regarding the relationships between classes. By using the global distribution information to reshape the logits of the preserved samples, the distributional shift of an incomplete dataset can be mitigated. We use Figure 4 to illustrate the effect of logit reshaping.

## 4   EXPERIMENTS

This section compares the effectiveness of different dataset pruning methods for distillation. Several teacher-student combinations are introduced to verify the generalization abilities of these methods on different datasets. Experimental details are listed as follows.

**Baselines.** Existing static dataset pruning methods such as Herding (Welling, 2009), Forgetting (Toneva et al., 2019), EL2N (Paul et al., 2021) and MoDS (Xia et al., 2023) are included for comparison. Different from these methods that are not designed for KD, Ben-Baruch et al. (2024) proposes to randomly prune samples for distillation and adaptively tune the hyper-parameters of the distillation loss according to the pruning ratio. We compare the method in (Ben-Baruch et al., 2024) in experiments, which is denoted by Adaptive. Following Adaptive, we randomly prune samples from the dataset in a class-balanced manner to form a baseline method Random. For a fair comparison, the proposed method also selects the medium-difficulty samples evenly from each class. In addition, a recently proposed dynamic pruning method InfoBatch (Qin et al., 2024) is also included.

**Datasets.** The main experiments are conducted on two benchmark datasets, i.e., CIFAR-100 (Krizhevsky et al., 2009) and ImageNet (Russakovsky et al., 2015). CIFAR-100 contains 60,000 images in total belonging to 100 classes. The dataset is divided into a training set with 50,000 images and a testing set with 10,000 images. ImageNet is a larger dataset collecting images from 1,000 classes. It contains 1,281,167 training images and 50,000 testing images. The images on CIFAR-100 and ImageNet are resized to $32 \times 32$ and $224 \times 224$ pixels, respectively.

**Teacher-student combinations.** We evaluate the generalization ability of different pruning methods by using different network combinations. The architectures of teacher networks include ResNet34 (He et al., 2016), ResNet50 (He et al., 2016), VGG13 (Simonyan & Zisserman, 2014), WRN-40-2 (Zagoruyko & Komodakis, 2016) and ResNet32x4 (He et al., 2016). The architectures of student networks include MobileNet (Howard et al., 2017), MobileNetV2 (Sandler et al., 2018), VGG8 (Simonyan & Zisserman, 2014), ResNet8x4 (He et al., 2016) and WRN-16-2 (Zagoruyko & Komodakis, 2016).

**Training setup.** We follow the training scheme of a popular open-source distillation repository (Tian et al., 2020) to set up the learning rate, learning rate decay rate, training epochs, batch size, weight decay rate and optimizer. On CIFAR-100, the hyper-parameter $\alpha$ is set to 0.1. For Adaptive, we set $\alpha = 0.1$, $\alpha = 0.15$, $\alpha = 0.2$ and $\alpha = 0.5$ when the keep ratio $r$ is 30%, 40%, 50% and 70%, respectively. On ImageNet, we set the hyper-parameters to 0.5 and 0.9 for the cross-entropy loss and the distillation loss, respectively. In addition, the temperature coefficient is set to 4 in all experiments. The results are obtained by using an NVIDIA V100 GPU.

## 4.1 Ablation Studies

We first investigate the distillation performance with samples selected from $d^H$, $d^E$, $d^M$. Table 1 lists the top-1 accuracy of different students distilled with different types of samples. The results on CIFAR-100 are averaged over 3 trials and the standard deviation is also reported. As shown in the table, since hard samples are difficult for the student to learn, the student distilled with hard samples underperforms those distilled with random samples in a low keep ratio. However, as the ratio increases, the situation is different. This is because the student

Table 1: Top-1 accuracy±standard deviation (%) on CIFAR-100 with ResNet50-MobileNetV2 using different data pruning strategies. Reshape denotes reshaping logits with the average information.

| Strategy | $r = 40\%$ | $r = 70\%$ |
|---|---|---|
| Random | 55.95±0.89 | 63.89±1.03 |
| Hard | 54.17±0.67 | 64.99±0.29 |
| Easy | 56.20±0.56 | 63.57±0.27 |
| Medium | 57.59±0.06 | 65.59±0.40 |
| Medium+Reshape | **58.04±0.74** | **66.18±0.11** |

has more training iterations to learn the teacher's feature distribution with a higher keep ratio. On the other hand, the teacher's knowledge of easy samples is easy to learn and helps the student obtain better performance in a low keep ratio compared to using random samples. However, as shown in Figures 1 and 2, the decision boundary formed by easy samples is limited and can only cover a small portion of features, which degrades the generalization ability of the student. By selecting medium-difficulty samples, the proposed method obtains a better trade-off between sample difficulty and the network's generalization ability achieving the best performance with different keep ratios of data. In addition, by using the logit reshaping technique, the student's accuracy can be further increased.

## 4.2 MORE JUSTIFICATION OF USING MEDIUM-DIFFICULTY SAMPLES

Apart from comparing the classification accuracy, we investigate the effects of using different types of samples for distillation from the perspectives of gradient variation and mutual information. Specifically, we compute the L2-norm of the gradient of the last layer by using different samples for training and obtain the standard deviation of the gradient magnitudes over different training epochs. In this way, we can analyze how different samples affect the network's learning. In addition, we measure the alignment between decision boundaries formed by the student and the teacher by using mutual information.

As shown in Table 2, the standard deviation of gradients of hard samples is lower, leading to a more stable training process. However, the mutual information of hard samples is also lower, which indicates the student's decision boundaries of hard samples do not well align with that of the teacher. Although the mutual information is high in the case of easy samples, the training process is unstable in terms of the standard deviation of gradient magnitudes, which may result in overfitting and affect the generalizability of the student. Overall, medium-difficulty samples obtain a trade-off between standard deviation and mutual information.

Table 2: Standard Deviation (SD) of gradients and Mutual Information (MI) of decision boundaries on CIFAR-100 with ResNet50-MobileNetV2 using different strategies.

| Strategy | SD | MI |
|---|---|---|
| Hard | 0.018 | 0.046 |
| Easy | 0.030 | 0.054 |
| Medium | 0.022 | 0.051 |

## 4.3 COMPARISON OF DIFFERENT LOGIT RESHAPING METHODS

The technique of logit reshaping has been well-studied in the existing KD methods. For instance, Spherical Knowledge Distillation (SKD) (Guo et al., 2020) proposes to align the magnitude of the student logit with that of the teacher logit. NormKD (Chi et al., 2023) rescale student logits and teacher logits with their corresponding standard deviations. Logit Standardization in Knowledge Distillation (LSKD) (Sun et al., 2024) further reshape the logit to a zero mean vector based on NormKD. The technical details of existing logit

Table 3: Top-1 accuracy±standard deviation (%) on CIFAR-100 with ResNet50-MobileNetV2 using different logit reshaping methods.

| Method | $r = 40\%$ | $r = 70\%$ |
|---|---|---|
| Medium | 57.59±0.06 | 65.59±0.40 |
| SKD | 56.40±0.41 | 64.94±0.27 |
| NormKD | 55.99±0.93 | 65.55±0.31 |
| LSKD | 56.25±0.91 | 65.06±0.50 |
| Ours | **58.04±0.74** | **66.18±0.11** |

reshaping methods are described in Appendix A.3. Different from these logit reshaping methods that focus on aligning the smoothness of student and teacher logits, we aim to align the distribution of the preserved teacher logits with the global distribution information for distillation on a pruned dataset. As shown in Table 3, existing logit reshaping methods degrade the distillation performance with different keep ratios. The proposed method improves the student's accuracy by a clear margin.

## 4.4 DISCUSSIONS ON CIFAR-100

**Logit distillation.** Figure 5 compares the proposed method with the existing static dataset pruning methods on CIFAR-100 with different keep ratios. In these experiments, ResNet50-MobileNetV2 and ResNet50-VGG8 are used to test the teacher and the student that have different network architectures and VGG13-VGG8 is used to evaluate the teacher and the student that has similar network architectures. More combinations are given in Appendix A.4. Experimental results show that existing methods, such as Herding, Forgetting, and EL2N underperform the other methods in most cases since they tend to select easy or hard samples for training. Random and Adaptive obtain better performance by approximately evenly selecting easy, medium and hard samples from datasets. By comparison, MoDS and the proposed method focus more on the medium-difficulty samples and achieve consistent improvements for different keep ratios. The differences between our method and MoDS are two-fold. Firstly, we introduce a logit reshaping method to convey the global information of the teacher's predictions to mitigate the gradient bias of a pruned dataset. Secondly, we select medium-difficulty samples in a class-balanced manner to avoid classification bias in the teacher. Therefore, our method consistently performs better than MoDS as shown in Figure 5.

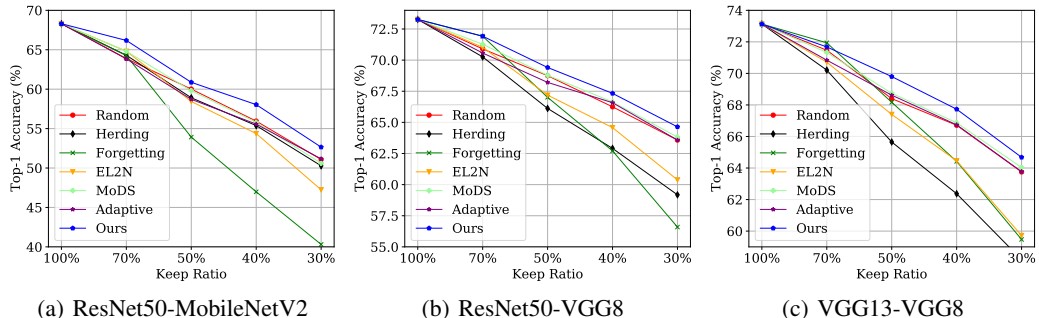

|                    |                    |                   |
| :----------------: | :----------------: | :---------------: |
| (a) ResNet50-MobileNetV2 | (b) ResNet50-VGG8 | (c) VGG13-VGG8 |

Figure 5: Top-1 accuracy of different static data pruning methods on CIFAR-100 datasets with different keep ratios. The proposed method consistently outperforms the existing pruning methods under different teacher-student combinations and network architectures.

Table 5: Top-1 accuracy (%) on ImageNet with different teacher-student pairs using different data pruning strategies. InfoBatch* is a dynamic pruning method, which needs to store the entire training set during distillation.

| Pair | ResNet34-MobileNet | | | ResNet50-MobileNet | | |
| :---: | :---: | :---: | :---: | :---: | :---: | :---: |
| Keep Ratio | $r = 30\%$ | $r = 50\%$ | $r = 70\%$ | $r = 30\%$ | $r = 50\%$ | $r = 70\%$ |
| Random | $67.25_{\downarrow 3.47}$ | $69.39_{\downarrow 1.33}$ | $70.20_{\downarrow 0.52}$ | $66.84_{\downarrow 3.81}$ | $68.96_{\downarrow 1.69}$ | $69.91_{\downarrow 0.74}$ |
| Herding | $61.96_{\downarrow 8.76}$ | $66.58_{\downarrow 4.14}$ | $68.88_{\downarrow 1.84}$ | $60.76_{\downarrow 9.89}$ | $66.30_{\downarrow 4.35}$ | $68.82_{\downarrow 1.83}$ |
| EL2N | $64.70_{\downarrow 6.02}$ | $69.14_{\downarrow 1.58}$ | $70.57_{\downarrow 0.15}$ | $63.47_{\downarrow 7.18}$ | $68.47_{\downarrow 2.18}$ | $70.16_{\downarrow 0.49}$ |
| Moderate | $66.79_{\downarrow 3.93}$ | $69.05_{\downarrow 1.67}$ | $69.93_{\downarrow 0.79}$ | $66.46_{\downarrow 4.19}$ | $68.72_{\downarrow 1.93}$ | $69.75_{\downarrow 0.90}$ |
| InfoBatch* | $67.43_{\downarrow 3.29}$ | $69.91_{\downarrow 0.81}$ | $70.47_{\downarrow 0.25}$ | $66.95_{\downarrow 3.70}$ | $69.83_{\downarrow 0.82}$ | $70.46_{\downarrow 0.19}$ |
| Ours | $\mathbf{67.91}_{\downarrow 2.81}$ | $\mathbf{69.98}_{\downarrow 0.74}$ | $\mathbf{70.92}_{\uparrow 0.20}$ | $\mathbf{67.67}_{\downarrow 2.98}$ | $\mathbf{69.97}_{\downarrow 0.68}$ | $\mathbf{70.93}_{\uparrow 0.28}$ |
| Full Dataset w/o KD | $69.57_{\downarrow 1.15}$ | | | $69.57_{\downarrow 1.08}$ | | |
| Full Dataset w/ KD | $70.72$ | | | $70.65$ | | |

**Feature distillation.** As a representative extension of the vanilla KD, feature distillation methods aim to distil the knowledge hidden in the intermediate layers of the teacher network. Table 4 evaluates the performance of different dataset pruning methods for feature distillation. In this experiment, we use the method in (Chen et al., 2022) to perform feature distillation. Similar to the results of logit distillation, selecting medium-difficulty samples

Table 4: Top-1 accuracy (%) on CIFAR-100 with ResNet50-MobileNetV2 using feature distillation.

| Strategy | $r = 40\%$ | $r = 70\%$ |
| :---: | :---: | :---: |
| Random | $57.68 \pm 0.95$ | $66.04 \pm 0.90$ |
| Herding | $55.42 \pm 0.40$ | $65.03 \pm 0.46$ |
| Forgetting | $50.41 \pm 0.33$ | $65.34 \pm 1.36$ |
| EL2N | $54.41 \pm 0.22$ | $65.95 \pm 0.54$ |
| MoDS | $57.61 \pm 0.25$ | $66.06 \pm 0.36$ |
| Ours (w/o Reshape) | $\mathbf{58.11 \pm 0.61}$ | $\mathbf{66.46 \pm 0.18}$ |

is beneficial to the feature distillation on a pruned dataset. The comparison between MoDS and the proposed method further demonstrates the importance of class-balanced sampling in distillation.

## 4.5 DISCUSSIONS ON IMAGENET

**Performance comparisons.** We further verify the effectiveness of the proposed method on a larger dataset ImageNet. Table 5 shows the top-1 accuracies of different methods with teacher-student pairs ResNet34-MobileNet and ResNet50-MobileNet. Experimental results show that the proposed method consistently outperforms the other static pruning methods under different ratios of preserved samples. In addition, the proposed method achieves lossless accuracy by solely using 70% of the training data. This is because the large-scale dataset contains more redundant samples. Therefore, it is necessary to develop effective dataset pruning methods to identify useful samples for training efficiency improvement. Apart from static pruning methods, a recently proposed dynamic pruning method InfoBatch is also included for comparison in Table 5. Since InfoBatch prunes different numbers of samples at different epochs, as in the original paper, we report the top-1 accuracies of

different keep ratios by matching the number of forward propagation during training. In addition, InfoBatch tends to preserve hard samples for training, which will also lead to the drift of the decision boundary. Although the proposed method does not access the entire dataset during training, it still obtains competitive performance compared to InfoBatch.

**Training complexity.** Since the proposed method directly utilizes the pre-trained teacher to rank the difficulties of training samples, the overhead for data selection can be significantly reduced. Therefore, we investigate how the pruned dataset affects the training time and accuracy of distillation. Figure 6 displays the curves of testing accuracy versus training time using the teacher-student pair ResNet34-MobileNet. By using 50% of training samples, the proposed method outperforms the student without using distillation by 0.41% in terms of top-1 accuracy. Besides, the training time of distillation can be halved by using dataset pruning. On the other hand, the student trained on a full dataset requires one and a half times of training time compared to the distillation using 50% of training data. In addition, by using the proposed method to facilitate logit distillation on the pruned datasets, the

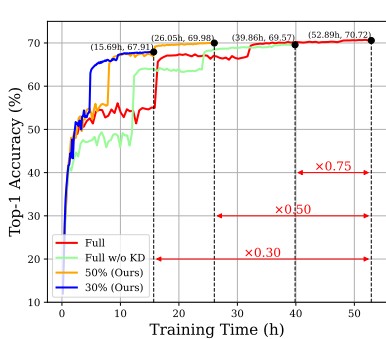

Figure 6: Top-1 accuracy and training time on ImageNet with different keep ratios.

distilled student converges faster with different keep ratios compared to the training on a full dataset.

**Comparisons with self-knowledge distillation.** Self-knowledge distillation is another technical route for accelerating the training of KD. Therefore, we compare our method with several recently proposed self-knowledge distillation methods in Table 6. The top-1 accuracies of Zipf's LS and USKD are cited from (Yang et al., 2023). Since

Table 6: Top-1 accuracy and training time on ImageNet of self-knowledge distillation and the proposed method with ResNet34-MobileNet using 70% of samples.

|          | Baseline | Zipf's LS | USKD   | Ours    |
|----------|----------|-----------|--------|---------|
| Acc (%)  | 69.57    | 69.59     | 70.38  | **70.92** |
| Time (h) | 39.86    | >39.86    | >39.86 | **36.98** |

the self-knowledge distillation methods need to use the entire dataset and additional distillation losses for training, their computation complexities will be slightly higher than the standard training procedure (i.e., Baseline). However, the proposed method can significantly reduce the training cost by using 70% of training data. In terms of the student's accuracy, the improvement of self-knowledge distillation methods is restricted due to the lack of the teacher's knowledge. By eliminating less important samples, the proposed method can speed up the training process and benefit from the supervision of the teacher. In addition, the performance of our method can be further improved by equipping with a more advanced distillation loss as shown in Appendix A.7.

## 5 CONCLUSION

This paper proposes a dataset pruning method with logit reshaping to accelerate the training of KD. Specifically, we observe that the teacher's knowledge of hard samples is difficult for the student to learn, which will lead to a deviation of the decision boundary. Therefore, we suggest preserving medium-difficulty samples for dataset pruning so that students can achieve smooth and discriminative decision boundaries between classes. In addition, to mitigate the gradient bias caused by dataset pruning, an improved distillation loss is proposed by utilizing the global information of the teacher's predictions to reshape the logits of the preserved samples. Experiments on different datasets with multiple teacher-student pairs verify the effectiveness and generalizability of the proposed method.

## 6 ACKNOWLEDGMENTS

This work is supported by Australian Research Council DP230101753 and CSIRO's Research Plus Science Leader Project R-91559.

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

# A APPENDIX

## A.1 IMPLEMENTATION DETAILS

Following CRD Tian et al. (2020), we set the batch size, training epochs, and weight decay rate to 64/256, 240/100, and 0.0005/0.0001 on CIFAR-100/ImageNet. The initial learning rate is 0.01 for MobileNetV2 and 0.05 for the other students and is multiplied by 0.1 at 150, 180, and 210 epochs on CIFAR-100. On ImageNet, the initial learning rate is 0.1 and is multiplied by 0.1 at 30, 60, and 90 epochs. In addition, the commonly used data augmentation techniques, e.g., random crop and horizontal flip are utilized for training. The optimizer is Stochastic Gradient Descent (SGD) with a momentum of 0.9. The feature distillation loss we use in Table 4 is as follows:

$$\mathcal{L}_{FD} = \frac{1}{2}||\frac{f(s)}{||f(s)||_2} - \frac{t}{||t||_2}||_2^2, \tag{6}$$

where $s$ and $t$ are the student and teacher feature vectors of a certain sample, respectively, and $f(s)$ is a function used to align the feature dimensions of the student and teacher with the projector ensemble mechanism. The hyper-parameters of the cross-entropy loss, the distillation loss and the number of projectors are set to 1, 25 and 3, respectively, as in the code released by the authors[2].

## A.2 DETAILS OF REPRODUCING BASELINES OF DATASET PRUNING

In the experiments, we reproduce existing dataset pruning methods by using the codes released by the authors or implement the codes according to the descriptions of the corresponding paper. The technical details of different baselines are as follows:

- Random (Ben-Baruch et al., 2024): We evenly prune samples from different classes to achieve class-balanced sampling. Ben-Baruch et al. (2024) demonstrates such a dataset pruning strategy can outperform existing methods in KD.

- Herding (Welling, 2009): We preserve samples with deeper features generated by the pre-trained teacher that are closer to the corresponding class centres.

- Forgetting (Toneva et al., 2019): We use the code released by the authors[3] to re-train the teacher network and monitor the forgetting events of samples. Samples that are harder to memorize (i.e., the number of forgetting events is higher) during training will be preserved.

- EL2N (Paul et al., 2021): The original EL2N compares the discrepancy between the ground-truth labels and the predictions of networks trained with a few epochs. In our experiments, we use the pre-trained teacher to compute the EL2N scores. Samples with larger scores are preserved for distillation.

- MoDS (Xia et al., 2023): We use the code released by the authors[4] to select samples. Specifically, samples with deeper features generated by the pre-trained teacher that are closer to the median distance of the corresponding class centres are preserved.

- Adaptive (Ben-Baruch et al., 2024): This baseline adopts the same dataset pruning strategy as Random.

- InfoBatch (Qin et al., 2024): We use the code released by the authors[5] to dynamically prune data during distillation. Samples that have larger $\mathcal{L}_{KD}$ values will be preserved at the next training epochs. In addition, InfoBatch re-scales the loss according to the data pruning ratio for further improvement.

## A.3 TECHNICAL DETAILS OF EXISTING LOGIT RESHAPING METHODS

Let $p$ and $z$ denote the teacher and the student logit vectors, the logit reshaping methods normalize the logit vectors as follows:

---

[2]https://github.com/chenyd7/PEFD
[3]https://github.com/mtoneva/example_forgetting
[4]https://github.com/tmllab/Moderate-DS
[5]https://github.com/NUS-HPC-AI-Lab/InfoBatch/tree/master

$$\textbf{SKD:} \; p = p, z = z\frac{\mathcal{M}_{STD}(p)}{\mathcal{M}_{STD}(z)} \tag{7}$$

$$\textbf{NormKD:} \; p = \frac{p}{\alpha\mathcal{M}_{STD}(p)}, z = \frac{z}{\alpha\mathcal{M}_{STD}(z)} \tag{8}$$

$$\textbf{LSKD:} \; p = \frac{p - \bar{p}}{\mathcal{M}_{STD}(p)}, z = \frac{z - \bar{z}}{\mathcal{M}_{STD}(z)} \tag{9}$$

$$\textbf{Ours:} \; p = p\frac{k_p}{\mathcal{M}_{STD}(p)}, z = z \tag{10}$$

where $\bar{p}$ is the mean of $p$, $\alpha$ is a hyper-parameter. Unlike the previous methods that aim to align the scale of the student logits with those of the teacher logits, the proposed method focuses on aligning the teacher logits with their corresponding class-wise information, mitigating the distribution shift due to dataset pruning. Therefore, the proposed method is significantly different from the previous logit reshaping methods. In addition, we can integrate these methods in one framework, i.e., $p = \beta_1 p$ and $z = \beta_2 z$, where $\beta_1$ and $\beta_2$ are different coefficients for scaling. To apply different logit reshaping methods, we just need to set the corresponding coefficients (e.g., $\beta_1 = k_p/\mathcal{M}_{STD}(p), \beta_2 = 1$).

### A.4 MORE COMBINATIONS ON CIFAR-100

In this subsection, we introduce more teacher-student combinations for evaluation as shown in Tables 7 to 10. Combinations WRN-40-2-WRN-16-2 and ResNet32x4-ResNet8x4 share similar network architectures while combinations WRN-40-2-VGG8 and ResNet32x4-VGG8 have different network architectures. In these experiments, we perform logit distillation between the student and the teacher with the keep ratios of samples ranging from $[30\%, 40\%, 50\%, 70\%]$. Experimental results in these tables demonstrate that the proposed method performs better than the other static dataset pruning methods in most cases. Overall, we evaluate the effectiveness of different methods with about nine different teacher-student pairs. We use the ratio of parameters of teacher-student pairs to roughly quantify their learning capability discrepancy (i.e., parameters of student network/parameters of teacher network). The ratios of these pairs range from $[0.02, 0.08, 0.16, 0.2, ..., 0.87]$, involving teacher and student networks with different learning capabilities. The diversity of the teacher-student pairs and the corresponding results demonstrate the generalizability of the proposed method.

Table 7: Top-1 accuracy±standard deviation (%) on CIFAR-100 dataset using a teacher-student pair with similar network architectures.

| Teacher-Student Pair | WRN-40-2-WRN-16-2 | | | | |
|---|---|---|---|---|---|
| Strategy / Ratio | $r = 30\%$ | $r = 40\%$ | $r = 50\%$ | $r = 70\%$ | Full Dataset |
| Random | 67.73±0.07 | 69.69±0.09 | 71.13±0.37 | 73.29±0.25 | |
| Herding | 61.93±0.25 | 65.52±0.24 | 68.37±0.26 | 71.99±0.15 | |
| EL2N | 64.79±0.59 | 68.23±0.39 | 70.53±0.05 | 73.12±0.12 | 74.82±0.21 |
| Moderate | 67.87±0.23 | 70.40±0.07 | 71.59±0.31 | 73.63±0.11 | |
| Adaptive | 67.73±0.07 | 70.04±0.19 | 71.11±0.30 | 72.98±0.09 | |
| Ours | **68.93±0.56** | **70.56±0.09** | **72.33±0.20** | **73.81±0.20** | |

Table 8: Top-1 accuracy±standard deviation (%) on CIFAR-100 dataset using a teacher-student pair with similar network architectures.

| Teacher-Student Pair | ResNet32x4-ResNet8x4 | | | | |
|---|---|---|---|---|---|
| Strategy / Ratio | $r = 30\%$ | $r = 40\%$ | $r = 50\%$ | $r = 70\%$ | Full Dataset |
| Random | 63.64±0.36 | 66.47±0.07 | 68.45±0.05 | 71.08±0.34 | |
| Herding | 59.69±0.16 | 63.46±0.21 | 66.58±0.04 | 70.57±0.21 | |
| EL2N | 61.71±0.27 | 65.65±0.37 | 68.48±0.17 | 71.37±0.13 | 73.24±0.23 |
| Moderate | **65.73±0.20** | 68.04±0.31 | 69.80±0.30 | 71.67±0.33 | |
| Adaptive | 63.64±0.36 | 66.46±0.08 | 68.24±0.22 | 70.92±0.15 | |
| Ours | 65.46±0.18 | **68.33±0.23** | **69.80±0.14** | **72.77±0.13** | |

Table 9: Top-1 accuracy±standard deviation (%) on CIFAR-100 dataset using a teacher-student pair with dissimilar network architectures.

| Teacher-Student Pair | ResNet32x4-VGG8 | | | | |
|---|---|---|---|---|---|
| Strategy / Ratio | $r = 30\%$ | $r = 40\%$ | $r = 50\%$ | $r = 70\%$ | Full Dataset |
| Random | 61.40±0.14 | 64.50±0.22 | 66.81±0.18 | 69.84±0.06 | |
| Herding | 57.91±0.25 | 62.13±0.40 | 65.42±0.40 | 69.68±0.12 | |
| EL2N | 58.74±0.48 | 63.10±0.17 | 66.04±0.48 | 69.75±0.42 | 72.50±0.19 |
| Moderate | 62.79±0.32 | 65.66±0.29 | 67.69±0.18 | 70.51±0.18 | |
| Adaptive | 61.40±0.14 | 64.15±0.17 | 66.30±0.32 | 69.54±0.33 | |
| Ours | **63.07±0.22** | **66.09±0.49** | **68.18±0.08** | **70.97±0.24** | |

Table 10: Top-1 accuracy±standard deviation (%) on CIFAR-100 dataset using a teacher-student pair with dissimilar network architectures.

| Teacher-Student Pair | WRN-40-2-VGG8 | | | | |
|---|---|---|---|---|---|
| Strategy / Ratio | $r = 30\%$ | $r = 40\%$ | $r = 50\%$ | $r = 70\%$ | Full Dataset |
| Random | 64.06±0.18 | 66.63±0.21 | 68.49±0.26 | 71.30±0.20 | |
| Herding | 59.52±0.08 | 63.55±0.24 | 66.70±0.08 | 70.67±0.23 | |
| EL2N | 60.08±0.34 | 64.49±0.23 | 67.28±0.24 | 70.56±0.42 | 73.06±0.43 |
| Moderate | 64.10±0.25 | 66.80±0.13 | 69.08±0.44 | 71.31±0.22 | |
| Adaptive | 64.06±0.18 | 66.61±0.33 | 68.29±0.32 | 70.63±0.08 | |
| Ours | **65.37±0.14** | **67.97±0.22** | **69.62±0.14** | **72.10±0.34** | |

## A.5 EVOLUTION OF STUDENT FEATURE SPACE

In Figure 2, we visualize the decision boundaries formed by different types of samples in the student feature space. In this subsection, we further investigate the change of decision boundaries during the distillation process. Figures 7, 8, and 9 illustrate the evolution of decision boundaries formed by hard, easy, and medium samples at 120, 160 and 240 epochs, respectively. For hard samples, it is shown that the problem of the drift of decision boundaries is more severe at the early training stage. This is because the teacher's knowledge of hard samples is too complex for the student to learn. On the other hand, the training for easy samples converges rapidly and the constructed decision boundaries avoid overlap between different classes. However, these decision boundaries are too small to cover the overall feature distribution. The medium-difficulty samples constitute smoothed decision boundaries during the distillation process, leading to a discriminative and generalizable student network.

## A.6 RESULTS ON A HIGHLY-IMBALANCED DATASET

To verify if the proposed logit reshaping method works in a highly imbalanced dataset, we sample the CIFAR-100 dataset in an imbalanced manner. To be specific, we set the sampling rates to be $[100\%, 99\%, 98\%, ..., 1\%]$ for class $[0, 1, 2, ...99]$, respectively, to create a highly imbalanced dataset and use this imbalanced dataset for the student's distillation. Experimental results in Table 11 demonstrate that even if the dataset is highly imbalanced, the proposed logit reshaping method still improves the student's performance in terms of results in the second and third columns. This further verifies the generalizability of the proposed method. In addition, there is a potential way to optimize the proposed method further. Specifically, instead of aligning the teacher's logits with the class-specific distributions, we align the teacher's logits with the average distribution information of the entire dataset (i.e., the mean of class-specific distributions). In this way, we can balance the logit variance between the minority and the majority classes. Results in the last column demonstrate the effectiveness of such a reshaping technique.

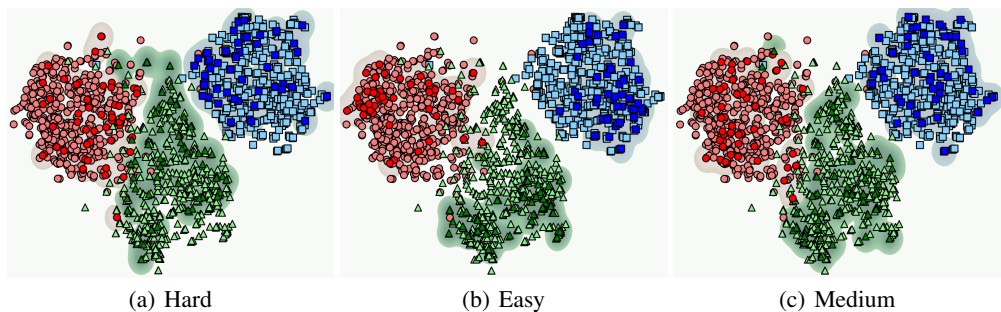

(a) Hard   (b) Easy   (c) Medium

Figure 7: Decision boundaries of (a) hard samples, (b) easy samples, and (c) medium samples selected by the teacher ResNet50 on the feature space of the distilled student MobileNetV2 on CIFAR-100 at **120** epochs.

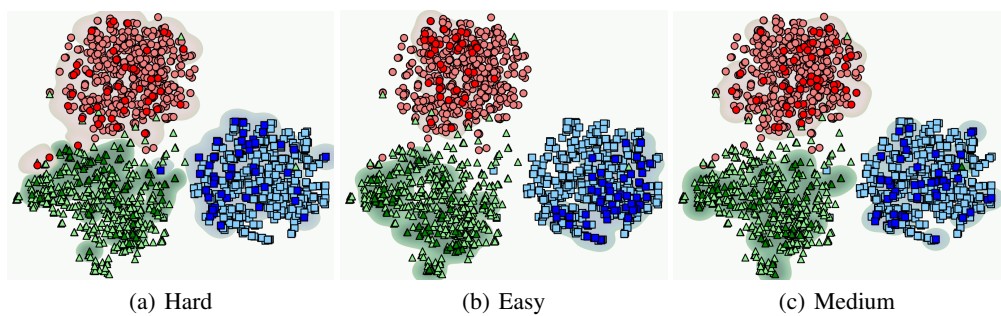

(a) Hard   (b) Easy   (c) Medium

Figure 8: Decision boundaries of (a) hard samples, (b) easy samples, and (c) medium samples selected by the teacher ResNet50 on the feature space of the distilled student MobileNetV2 on CIFAR-100 at **160** epochs.

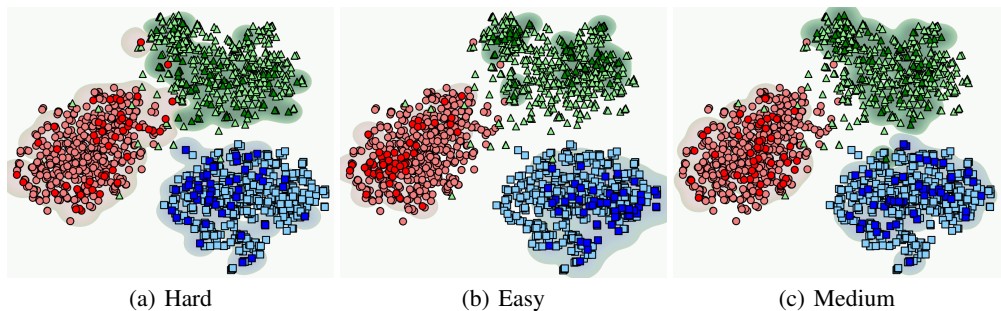

(a) Hard   (b) Easy   (c) Medium

Figure 9: Decision boundaries of (a) hard samples, (b) easy samples, and (c) medium samples selected by the teacher ResNet50 on the feature space of the distilled student MobileNetV2 on CIFAR-100 at **240** epochs.

## A.7 ADVANCED DISTILLATION LOSS

As mentioned in the Introduction, various extensions of the vanilla KD have been proposed in recent years. Since this paper focuses on the acceleration of distillation, we mainly adopt the vanilla KD for illustration in the experiments. This subsection demonstrates that the student's accuracy can be further increased by combining the proposed method with an advanced logit distillation loss. Specifically, the recently proposed Decoupled Knowledge Distillation (DKD) Zhao et al. (2022) firstly decomposes the vanilla logit distillation loss into a Target Class Knowledge Distillation (TCKD)

Table 11: Experiments on CIFAR-100 with imbalanced sampling by using ResNet50-MobileNetV2.

| Method | w/o Reshaping | w/ Reshaping | w/ Improved Reshaping |
|---|---|---|---|
| Top-1 Testing Accuracy (%) | 56.64 | 57.46 | 58.09 |

Table 12: Top-1 accuracy (%) on ImageNet by using different logit distillation losses with the proposed dataset pruning and logit reshaping method. The student network is MobileNet.

| Teacher | ResNet34 | | ResNet50 | |
|---|---|---|---|---|
| Method | $r = 50\%$ | $r = 70\%$ | $r = 50\%$ | $r = 70\%$ |
| KD | 69.98 | 70.92 | 69.97 | 70.93 |
| DKD | **70.17** | **71.27** | **70.22** | **71.61** |

loss and a Non-target Class Knowledge Distillation (NCKD) loss and the applies different hyperparameters for these two losses. Following the code released by the authors[6], we set the hyperparameters of TCKD and NCKD to be 1 and 0.5 on ImageNet, respectively. As shown in Table 12, the student's performance can be consistently improved by using DKD under different keep ratios, which demonstrates the generalizability of the proposed method.

## A.8 TRANSFER LEARNING

We further validate the performance of the distilled students from the view of transfer learning. In these experiments, we introduce two image datasets CUB-200 (Wah et al., 2011) and Cars-196 (Krause et al., 2013) for evaluation. For classification on a different dataset, we freeze the parameters of the backbones of the students distilled on ImageNet and re-train the last classification layer. Table 13 lists the top-5 accuracy of the students distilled with different static dataset pruning methods. As shown in the table, the proposed method outperforms the existing dataset pruning methods by a clear margin on different datasets.

Table 13: Top-5 accuracy (%) on CUB-200-2011 and Cars-196 datasets using the pre-trained student MobileNet distilled by different static dataset pruning strategies with keep ratio $r = 30\%$.

| Teacher | ResNet34 | | ResNet50 | |
|---|---|---|---|---|
| Strategy | CUB200 | Cars196 | CUB200 | Cars196 |
| Random | 88.59 | 76.65 | 88.02 | 74.41 |
| Herding | 87.78 | 75.35 | 87.02 | 72.35 |
| EL2N | 87.19 | 74.75 | 85.76 | 71.54 |
| Moderate | 89.04 | 75.11 | 87.81 | 73.62 |
| Ours | **89.48** | **78.10** | **88.31** | **75.97** |

## A.9 VISUALISATION OF SELECTED SAMPLES

To illustrate the differences between the preserved images, we visualize some images on ImageNet selected by the pre-trained teacher ResNet50. Figure 10 displays the images of hard samples from three different classes. It is shown that images of hard samples are more difficult to learn since these images contain more uncorrelated objects as in the image of "Ballon" or fewer features of the object as in the image of "Red wolf". In addition, hard samples may contain noise as the image of "Hamster". On the contrary, images of easy samples are easy to distinguish as shown in Figure 11. From these visualizations, medium-difficulty samples achieve a trade-off between easy and hard samples as shown in Figure 12 so that the student can learn accurate and generalizable decision boundaries between classes.

---

[6]https://github.com/megvii-research/mdistiller

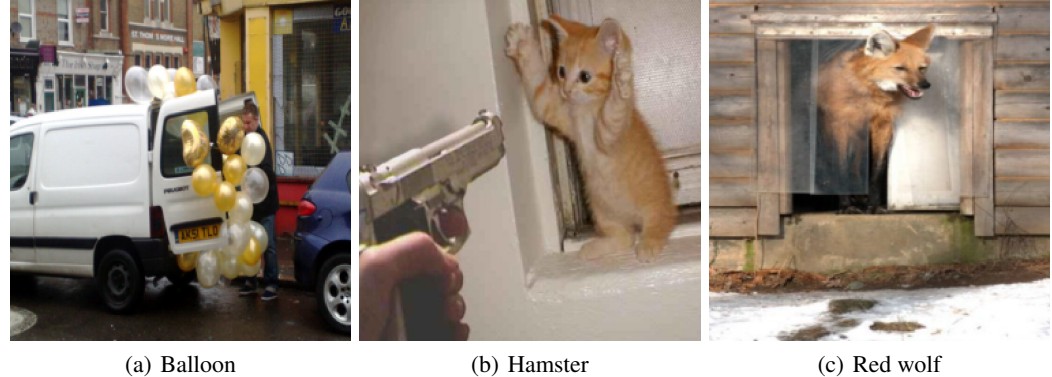

(a) Balloon                    (b) Hamster                    (c) Red wolf

Figure 10: Images of hard samples selected by the pre-trained ResNet50 on ImageNet.

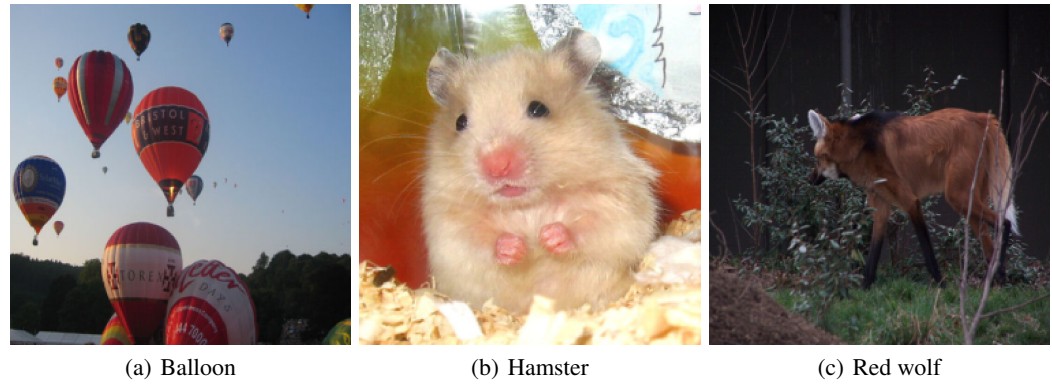

(a) Balloon                    (b) Hamster                    (c) Red wolf

Figure 11: Images of easy samples selected by the pre-trained ResNet50 on ImageNet.

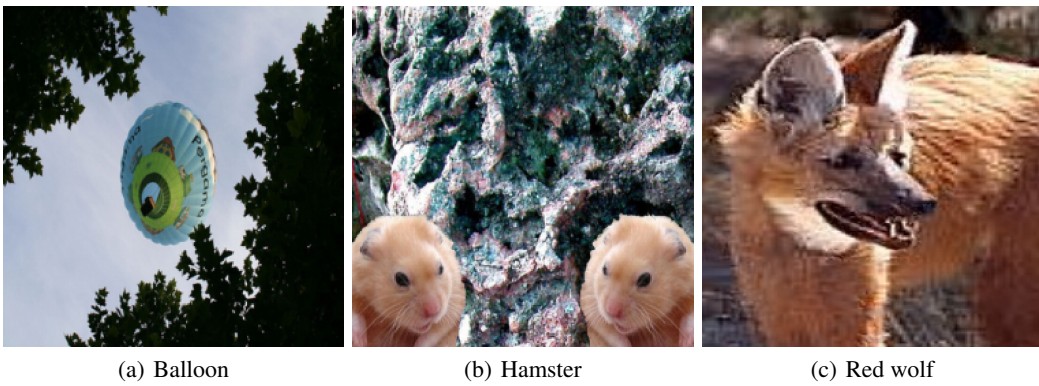

(a) Balloon                    (b) Hamster                    (c) Red wolf

Figure 12: Images of medium samples selected by the pre-trained ResNet50 on ImageNet.

