# OpenReview forum: "Medium-Difficulty Samples Constitute Smoothed Decision Boundary for Knowledge Distillation on Pruned Datasets"
_ICLR.cc/2025/Conference — ICLR 2025 Poster_

### Official Review · Reviewer_EGEk · 2024-10-31

**Soundness:** 3
**Presentation:** 3
**Contribution:** 3
**Rating:** 8
**Confidence:** 5

**Summary:**

The paper deals in data pruning for knowledge distillation. While previous observations show that random pruning is superior over other pruning methods, the authors proposed to retain medium difficulty samples for distillation. The motivation is that these samples are closer to the median distances between class centers.

**Strengths:**

- The explanation the authors suggest for the question of why a student model is sub-optimal when performing hard-sample based pruning methods is interesting and looks like it is the main novelty of the paper.  Also, The visualization in Figure 3 is clear and insightful.

- Experimental results are encouraging and look promising.

**Weaknesses:**

- In Figure  2 the authors show “a large accuracy gap” between the teacher and the student. The figure which indeed shows “Large accuracy gap” does not clearly demonstrate the specific degradation of the student’s accuracy due to the dataset pruning used. An accuracy gap between teacher and student is what we typically see in knowledge distillation in general. What does it mean “large accuracy gap”? Maybe the authors better show accuracy plots for easy, medium and hard pruning. Currently, Figure 2 is out of context.


- Equation (3), the M_DIF is computed to score hard samples. It is not quit clear why the M_DIF is defined here and what is the context? Is it another form of data pruning algorithm? (in figure 5, various data pruning are compared by the authors). Or it severs only the discussion in section 3.2? I would clarify the motivation.

**Questions:**

Please see the weaknesses.

---

> ### Author Response · Authors · 2024-11-21
> **Response to Reviewer EGEk**
>
> We are thankful for the reviewer's time in reviewing our paper.
>
> **Q1: In Figure 2 the authors show “a large accuracy gap” between the teacher and the student. The figure which indeed shows “Large accuracy gap” does not clearly demonstrate the specific degradation of the student’s accuracy due to the dataset pruning used. An accuracy gap between teacher and student is what we typically see in knowledge distillation in general. What does it mean “large accuracy gap”? Maybe the authors better show accuracy plots for easy, medium and hard pruning. Currently, Figure 2 is out of context.**
>
> A1: Figure 2 is used to illustrate that the student is unable to fully mimic the teacher's behaviour during the distillation process, which motivates us to investigate the discrepancy between the decision boundary formed by the student and the teacher. We apologize for the unclear explanations. Following the reviewer's comments, we take a closer look at the accuracies of different training samples obtained by the student. Specifically, we divide the training dataset into three subsets. Each subset corresponds to 30\% of hard samples, 30\% of medium samples, and 30\% of easy samples selected by the teacher, respectively. After training the teacher and distilling the student with the full dataset, the detailed top-1 training accuracy (\%) of each subset is as follows:
>
> |     | Hard |Medium |Easy  |All |
> |-------------|----------|----------|----------|----------|
> | Teacher |99.80 |99.92 |99.97|99.91|
> |Student |70.57 |85.21 |94.42|83.69|
>
> As shown in the above table, the accuracy gaps between samples of different difficulties are minor for the teacher network. However, it is a different case for the student network. Compared to the easy and medium samples, the hard samples tend to be misclassified in the student space. This phenomenon indicates that simply selecting hard samples for distillation may lead to the underfitting of the student and result in a degraded decision boundary for the student network. Such hypothesis is further verified by the visualization of decision boundaries formed by different samples as shown in Figure 3 in the manuscript. We thank the reviewer for this suggestion and will add the above results in the revised manuscript.
>
> **Q2: Equation (3), the $M_{DIF}$ is computed to score hard samples. It is not quit clear why the $M_{DIF}$ is defined here and what is the context? Is it another form of dataset pruning algorithm? (in figure 5, various dataset pruning are compared by the authors). Or it severs only the discussion in section 3.2? I would clarify the motivation.**
>
> A2: We thank the reviewer for this important advice. Existing dataset pruning methods score the difficulties of training samples by using different measurements. For example, EL2N, which is a baseline in our experiment, obtains the score of a training sample by computing the L2 distance between the network's prediction and the one-hot label vector. Our scoring mechanism is the same as that of EL2N except that we use the cross-entropy loss instead of the L2 distance since we conduct experiments on image classification task, which generally adopts the cross-entropy loss for optimization. We will clarify this in the revised manuscript for better readability.

---

### Official Review · Reviewer_s3Pu · 2024-11-02

**Soundness:** 2
**Presentation:** 3
**Contribution:** 2
**Rating:** 3
**Confidence:** 5

**Summary:**

The paper aims to investigate experimentally how to select a balanced subset of a training set, based on a pretrained teacher, so as to reduce the training time needed to train students in the knowledge distillation (KD) framework while controlling the performance degradation of distilled students as much as possible, where the distilled students are trained on the balanced set. The proposed method consists of two phases. In phase 1, all samples in the original whole training set are sorted in the decreasing order of their corresponding cross entropy losses given by the pretrained teacher, those located in the middle range are selected, and the average of the standard deviation of a logit vector over all logit vectors within each class is computed and recorded. In phase 2 (corresponding to the training stage in KD), the logit vector of each selected sample from the pretrained teacher is first reshaped by the recorded average standard deviation of the class that the selected sample belongs to and  then passed to the KD objective function. To maintain balance, the sorting and sample selection can be carried out on a per class basis, as implied in the first paragraph of Section 4.3. Experiments were carried out on CIFAR 100 and ImageNet for several teacher-student pairs to verify the effectiveness and generalizability of the proposed method.

**Strengths:**

1. The paper is generally well written.

2. The methodology is easy to follow and simple to implement.

**Weaknesses:**

1. The value proposition in terms of the accuracy vs time savings is kind of weak in this line of research. First, the intended time savings is for the KD training phase, not for the inference phase. Time savings for the inference phase is more important. Second, there is always accuracy degradation when the training dataset is pruned. When the ratio of kept sample is high, the training time savings is small. To have a meaningful training time savings, the ratio of kept sample has to be much smaller, which, however, results in significant, unacceptable accuracy performance degradation.

2. In comparison with Moderate Dataset Selection (MoDS) and reshaping techniques proposed in the literature, the novelty of the proposed method is kind of limited.

3. Experiments are insufficient to demonstrate the universality of the proposed method with respect to different students with variety of learning capability. The proposed method is mainly motivated by the learning gap between the pretrained teacher and students. What would happen if the learning capability of the student is close to, or very far away from that of the pretrained teacher? If the pruned training dataset works only for certain students, but not for others, the method won't fly since finding which student fits by experiments also takes significant amounts of training time.

4. Experiments are insufficient to demonstrate its competitiveness with another technique called data distillation.

**Questions:**

1. How are the shaping methods in the literature applied in the setup for Table 2? The comparison in Table 2 seems  like apple vs orange, and may be not fair.

2. More experiments are needed to show the value of the proposed method.

3. The few shot cases considered in the paper ``BAYES CONDITIONAL DISTRIBUTION ESTIMATION FOR KNOWLEDGE DISTILLATION BASED ON CONDITIONAL MUTUAL INFORMATION'' by L. Ye (ICLR 2024) can also be used to reduce the KD training time. Please compare the performance of the proposed method with the few shot case in that paper in terms of accuracy vs training time to see if the proposed method in this paper is orthogonal.

---

> ### Author Response · Authors · 2024-11-21
> **Response to Reviewer s3Pu (part 1)**
>
> We are thankful for the reviewer's time in reviewing our paper.
>
> **Q1: The value proposition in terms of the accuracy vs time savings is kind of weak in this line of research.**
>
> A1: We respectively disagree with the reviewer about the value of dataset pruning. First, there are some cases in which time savings for the training phase are more important. For instance, in the early stages of testing ideas during research or developing proofs of concept, quick iterations are more valuable than achieving perfect accuracy. Therefore, it is necessary to select a representative subset from the training set to verify the effectiveness of the idea. Second, there are many influential papers (e.g., Forgetting (ICLR), EL2N (NeurIPS)) that proposed pruning the training dataset with the trade-off between accuracy and time savings, which demonstrate the significance of the dataset pruning technique and its necessity to the research community. In addition, the baseline methods in our experiments, such as Zipf's LS (ECCV) and USKD (ICCV), also aim to reduce the training costs of KD. These methods all encounter the problem of trade-off between accuracy and training time. However, their efforts are widely recognized by the top-tier conferences.
>
> On the other hand, as shown in Table 4 in the manuscript, the proposed method is able to save about 30\% of the training time of KD without compromising any performance. As training a deep model in the real world is resource-intensive, time-consuming and expensive, we believe that such a pruning ratio is meaningful and may give us some really good solutions for real-world use as commented in the strengths of reviewer RWDY and reviewer FjLn.
>
> **Q2: In comparison with Moderate Dataset Selection (MoDS) and reshaping techniques proposed in the literature, the novelty of the proposed method is kind of limited.**
>
> A2: First, from the dataset pruning strategy perspective, we would like to clarify that this paper does not aim to propose a sophisticated dataset pruning strategy as the existing methods are proven to be effective in many scenarios and easy to implement. Instead, we focus on the challenges for these methods to be applied in a specific but important scenario (i.e., knowledge distillation) and how they can be further improved for such scenarios, from a novel perspective of decision boundary drift. We show that the medium-difficulty samples constitute a smoothed decision boundary for distillation on pruned datasets and avoid the drift of the decision boundary. The phenomenon is unexplored in MoDS and the other reviewers consistently recognize the insight.
>
> Second, regarding the novelty of the proposed logit reshaping method, existing reshaping methods all fail to improve the student's performance on a pruned dataset as shown in Table 2 in the manuscript. By contrast, our method proposes a simple modification by recording the teacher's global information and successfully improves the student's performance. The results demonstrate the intrinsic discrepancy between the existing logit reshaping methods and the proposed method, which is not supposed to be considered a limited contribution. The technical details of different logit reshaping methods are listed in the answer for Q5.

---

> ### Author Response · Authors · 2024-11-21
> **Response to Reviewer s3Pu (part 2)**
>
> **Q3: Experiments are insufficient to demonstrate the universality of the proposed method with respect to different students with variety of learning capability.**
>
> A3: We thank the reviewer for the comment. First, to address the reviewer's concern, we conduct a distillation experiment using a teacher-student pair ResNet50-ResNet50. In this case, the teacher and the student have the same learning capability. We can find that selecting medium-difficulty samples is still better in most cases in terms of top-1 testing accuracy (\%) as follows:
>
> | Ratio    | 30\% |40\%  |50\%  |
> |-------------|----------|----------|----------|
> | Hard  |69.02  |73.14  |**76.26**  |
> |Easy |68.83 |72.00 |74.55      |
> |Medium |**69.70** |**74.00** |**76.09**|
>
> Results in the above table demonstrate the effectiveness of using medium-difficulty samples even when the student and the teacher have the same learning capability. We hypothesize that this is because the total training iterations are reduced due to the lack of a large amount of samples. Therefore, even if the student and the teacher have the same architectures, it is still challenging for the student to obtain the same local optimum as the teacher, leading to the drift of decision boundaries with hard and easy samples.
>
> Second, in our experiment, we evaluate the effectiveness of different methods with about ten different teacher-student pairs (including the above ResNet50-ResNet50). We use the ratio of parameters of teacher-student pairs to roughly quantify their learning capability discrepancy (i.e., parameters of student network/parameters of teacher network). The ratios of these pairs are ranged from [0.02, 0.08, 0.16, 0.2,..., 0.87, 1]. These pairs are diverse enough in terms of the ratios, involving teacher and student networks with different learning capabilities. In addition, these teacher-student combinations have been widely adopted as benchmarks in the existing KD paper. Therefore, we believe that our experiments are sufficient to verify the universality of the proposed method.
>
> **Q4: Experiments are insufficient to demonstrate its competitiveness with another technique called data distillation.**
>
> A4: We kindly emphasize that this paper aims to provide insights for practitioners that directly use the pre-trained models (e.g., downloaded from the Pytorch repository) to distil a student and prune the dataset using the prior knowledge from the pre-trained teacher without introducing too much computation overhead. Suppose we use the technique of data distillation, it requires additional training and will inevitably increase the overall training costs. Such computation overheads for data distillation may be higher than training a student network without distillation. Unlike data distillation, our method (dataset pruning) can improve the student's performance even with fewer training times as shown in Table 5 in the manuscript. Therefore, we argue that dataset pruning and data distillation are two different technical routes and comparing them is like apple vs orange, and is, indeed, unfair.
>
> **Q5: How are the shaping methods in the literature applied in the setup for Table 2? The comparison in Table 2 seems like apple vs orange, and may be not fair.**
>
> A5: Let $p$ and $z$ denote the teacher and the student logit vectors, the logit reshaping methods normalize the logit vectors as follows:
>
> SKD: $p=p, z=z\frac{std(p)}{std(z)}$
>
> NormKD: $p=\frac{p}{\alpha std(p)}, z=\frac{z}{\alpha std(z)}$
>
> LSKD: $p=\frac{p-\tilde{p}}{std(p)}, z=\frac{z-\tilde{z}}{std(z)}$
>
> Ours: $ p=p\frac{avgstd(p)}{std(p)}, z=z$
>
> where $\tilde{p}$ is the mean of $p$, $\alpha $ is a hyper-parameter, $std(p)$ is the standard deviation of $p$, and $avgstd(p)$ is the average standard deviation of the class that $p$ belongs to. Unlike the previous methods that aim to align the scale of the student logits with those of the teacher logits, the proposed method focuses on aligning the teacher logits with their corresponding class-wise information, mitigating the distribution shift due to dataset pruning. Therefore, the proposed method is significantly different from the previous logit reshaping methods. In addition, we can integrate these methods in one framework, i.e., $p=\beta_1 p$ and $z=\beta_2 z$, where $\beta_1$ and $\beta_2$ are different coefficients for scaling. To apply different logit reshaping methods, we just need to set the corresponding coefficients (e.g., $1/std(p)$). Therefore, they are comparable.

---

> ### Author Response · Authors · 2024-11-21
> **Response to Reviewer s3Pu (part 3)**
>
> **Q6: The few shot cases considered in the paper ... to see if the proposed method in this paper is orthogonal.**
>
> A6: We thank the reviewer for recommending the relevant paper. First, we struggle to reproduce the results reported in the paper. There are some errors and some files are missing in this repository. For example, the setting of hyper-parameters in file "runKD0.sh" is different from that of the descriptions in the paper. The command on how to fine-tune the pre-trained teacher and the file 'get\_cifar100\_sub\_dataloaders' is missing. Eventually, we tried to rerun the code according to the descriptions in the paper and there is still a discrepancy from the original results. Similar to the observation in the original paper, our reproduced results show that fine-tuning the teacher network by MCMI significantly improves the distillation performance in low sampling ratios (e.g., 25\% and 35\%) in terms of top-1 testing accuracy (\%) as follows:
>
> |    | 25\% |35\% |50\% |75\% |
> |-------------|----------|----------|----------|----------|
> | MLL    |64.35  |66.55   |68.04  |70.22         |
> |   MCMI   |65.27   |67.18  |68.20  |69.85        |
> |   MLL+Ours   |64.54   |66.61  |68.65  |70.23  |
>
> However, in high sampling ratios (e.g., 75\%), MCMI may degrade the distillation performance. Although the proposed method underperforms MCMI in low sampling ratios, it can consistently improve the student under different sampling ratios and does not need extra training costs to fine-tune the teacher network. Overall, we are interested in the promising performance that MCMI obtains in low sampling ratios and it is worth investigating how to select samples according to the teacher fine-tuned by MCMI. In summary, MCMI and our paper are orthogonal as MCMI focuses on learning a suitable teacher network for the student while our method aims to select appropriate training samples for distillation. We will cite this paper in the revised manuscript and thanks for the reviewer's recommendation.

---

> > ### Comment · Reviewer_s3Pu · 2024-11-28
> >
> > Thank you for your responses. My concerns expressed in the weakness section remain the same:
> >
> > 1. In Table 4, with the ratio of kept samples at 30%, there is around 3% performance degradation in comparison with KD based on the full training dataset, which is quite significant deterioration.
> >
> > 2. In Table 2, with the ratio of kept samples at 40%, it is really hard to tell which one is better between yours and the Medium method since the deviation in your case is much larger.
> >
> > 3. To address the universality concerns and demonstrate any potential value, you need to run experiments with different pairs of teacher and student, where the teacher is fixed, but the student varies from small architectures to large architectures. Following the standard KD setups won't achieve this purpose.

---

> ### Author Response · Authors · 2024-11-29
> **Response to Reviewer s3Pu (Part 4)**
>
> We thank the reviewer for the response. Note that the order of tables is different as we have submitted a revised manuscript.
>
> **Q1: In Table 4, with the ratio of kept samples at 30\%, there is around 3\% performance degradation in comparison with KD based on the full training dataset, which is quite a significant deterioration.**
>
> A1: We assume that the reviewer is referring to Table 5 in the revised manuscript. First, we respectively disagree with the reviewer about the performance degradation as it is unrealistic to expect an unchanged network's performance or almost the same performance when being trained on only 30\% of training samples. That being said, as mentioned in our previous response (i.e., A1 in "Response to Reviewer s3Pu (part 1)"), there are some cases in which time savings for the training phase are more important than achieving perfect performance (e.g., reducing costs for individuals and institutes with limited computing resources for the democratization of AI [1], hyper-parameter tuning [2], searching the optimal neural architecture [3],[4]).
>
> Second, we kindly emphasize that the relative improvement obtained by the proposed method is significant compared to the existing dataset pruning methods with a 30\% keep ratio. Specifically, with ResNet50-MobileNet, the top-1 accuracy of our method is **67.67\%** while the top-1 accuracies of the compared methods range from **[60.76\%, 66.95\%]**. With ResNet34-MobileNet, the top-1 accuracy of our method is **67.91\%** while the top-1 accuracies of the compared methods range from **[61.96\%, 67.43\%]**. Accuracy gains over 0.5\% are generally considered significant in the scenarios of data pruning [2] and KD [5,6].
>
> [1] Mansheej Paul et al. Deep Learning on a Data Diet: Finding Important Examples Early in Training, NeurIPS 2021.
>
> [2] Xiaobo Xia et al. Moderate Coreset: A Universal Method of Data Selection for Real-world Data-efficient Deep Learning, ICLR 2023.
>
> [3] Chongjun Tu et al. Efficient Architecture Search via Bi-level Data Pruning, 2023.
>
> [4] Emanuel Ben-Baruch et al. Distilling the Knowledge in Data Pruning, 2024.
>
> [5] Zhendong Yang et al. Masked Generative Distillation, ECCV 2022.
>
> [6] Linfeng Ye et al. Bayes Conditional Distribution Estimation for Knowledge Distillation Based on Conditional Mutual Information, ICLR 2024.
>
> **Q2: In Table 2, with the ratio of kept samples at 40\%, it is really hard to tell which one is better between yours and the Medium method since the deviation in your case is much larger.**
>
> A2: We assume that the reviewer is referring to Table 3 in the revised manuscript. As the number of training samples decreases significantly on CIFAR-100 with a 40\% keep ratio, the network's training may exhibit a larger variance in performance in this case, this is a normal phenomenon and it persists in all recent works including SKD, NormKD, LSKD, and ours. That is why we report the average accuracies of different methods over a few repetitions and conduct ablation studies with different keep ratios for a fair comparison. Nonetheless, if we assume the reported accuracy values for each model under each keep ratio to be following normal distributions, our method (1) outperforms Medium by 0.45\% by expected value with 40\% keep ratio, (2) outperforms Medium with 72.78\% probability by calculating the CDF of $P(X_2-X_1>0)$, where $X_1 \sim N(\mu_1 = 57.59, \sigma_1 = 0.06)$ and $X_2 \sim N(\mu_2 = 58.04, \sigma_2 = 0.74)$, with 40\% keep ratio. In addition, the proposed method achieves both better average accuracy and a smaller standard deviation with a 70\% keep ratio.
>
> We argue that the proposed method still demonstrates a clear advantage over other baseline methods. We also believe it is more appropriate to evaluate the performance of a method holistically rather than focusing on one specific number in the performance metrics, even though that number itself still demonstrates an advantage.

---

> ### Author Response · Authors · 2024-11-29
> **Response to Reviewer s3Pu (Part 5)**
>
> **Q3: To address the universality concerns and demonstrate any potential value, you need to run experiments with different pairs of teacher and student, where the teacher is fixed, but the student varies from small architectures to large architectures. Following the standard KD setups won't achieve this purpose.**
>
> A3: We would like to point out that **the requested information has been provided in the original submission and the previous response.** We did conduct experiments with different teacher-student pairs, where the teacher is fixed and the student varies from small architectures to large architectures. In addition, we also provided results from different teacher networks and their respective sets of students. Specifically, as shown in Figure 5 and Tables 7 to 10,  we have a fixed teacher network ResNet50 (23.7M Params), performing distillation for student networks MobileNetV2 (0.8M Params), VGG8 (3.9M Params), and ResNet50 (23.7M Params). And we also have another fixed teacher network WRN-40-2 (2.2M Params), performing distillation for student networks WRN-16-2 (0.7M Params), and VGG8 (3.9M Params).
>
> We appreciate any additional feedback and are happy to incorporate suggestions that ensure the quality of our work.

---

### Official Review · Reviewer_RWDY · 2024-11-02

**Soundness:** 2
**Presentation:** 2
**Contribution:** 1
**Rating:** 6
**Confidence:** 4

**Summary:**

In the paper, the authors addresses the challenge of efficient knowledge distillation (KD) on pruned datasets. The authors propose a static pruning method that focuses on selecting medium-difficulty samples, as opposed to traditional methods that prefer hard or easy samples. The method aims to mitigate decision boundary drift caused by the student's limited capacity to mimic the teacher's feature distribution. Additionally, the paper introduces a logit reshaping technique to reduce distributional shift in the pruned dataset by leveraging the teacher's global logit information. Experimental results demonstrate that the proposed method outperforms existing dynamic and static pruning techniques across various benchmarks, achieving notable efficiency and performance gains.

**Strengths:**

1. **Clear Motivation**: The authors clearly articulate the problem of decision boundary drift and how hard or easy samples influence KD differently. This sets a strong foundation for their proposed solution.
2. **Methodical Innovation**: The selection of medium-difficulty samples for KD and the use of global logit reshaping are novel concepts. The visualizations of decision boundaries and empirical evidence support the validity of these innovations.
3. **Comprehensive Evaluation**: The paper presents results from multiple datasets (CIFAR-100, ImageNet) and various teacher-student configurations. Ablation studies and comparisons with other pruning methods provide a holistic understanding of the approach’s effectiveness.
4. **Efficiency Gains**: By achieving better student performance while significantly reducing training times, the proposed method demonstrates practical utility, especially for resource-constrained settings.

**Weaknesses:**

1. **Theoretical Justification of Medium-Difficulty Samples**:
Despite the novelty of using medium-difficulty samples, the overall methodological advances feel incremental. The authors should need more rigorous theoretical analysis to make the proposed methodology convince to the reader of the underlying mechanisms driving performance improvements.

   - **Explicit Analysis**: The authors should provide a rigorous mathematical analysis that formalizes how medium-difficulty samples contribute to smoother decision boundaries. This could involve deriving relationships between sample difficulty, gradient stability, and boundary smoothness.
   - **Statistical Modeling**: Conduct an empirical risk analysis or use statistical measures to model the impact of medium-difficulty samples on decision boundary drift. For example, the authors could analyze the variance of gradients during training for different difficulty levels and quantify how medium-difficulty samples stabilize learning.
   - **Geometric Insights**: The authors can use tools from computational geometry to offer insights into how medium-difficulty samples influence class separability. Visualizing the feature space evolution with respect to class overlap during training could also strengthen their argument.

2. **Deeper Analysis of Logit Reshaping**:
   - **Impact Study on Logit Distribution**: Perform a sensitivity analysis on the logit reshaping process. Specifically, the authors can quantify how reshaping logits using the teacher’s global information affects the student’s performance across classes, especially in cases of class imbalance or noisy labels.
   - **Ablation of Reshaping Components**: Break down the logit reshaping mechanism and assess the impact of each component (e.g., the standard deviation adjustment and the mean logit alignment) to isolate what contributes most to performance gains.
   - **Bias and Overfitting Analysis**: Evaluate whether the logit reshaping method introduces bias or overfitting. This could involve testing the method on datasets with varying distributions and comparing generalization performance on unseen data.

3. **Quantitative Analysis of Decision Boundary Dynamics**:
   - **Boundary Drift Measurement**: The authors can introduce or provide a metric or framework for quantifying decision boundary drift in the student network. This could involve computing the alignment of decision boundaries between the teacher and student models using tools like mutual information or boundary discrepancy measures.
   - **Difficulty Spectrum Analysis**: Assess how the performance changes as samples are divided across a spectrum of difficulties, rather than just hard, medium, or easy categories. This can help establish a more granular understanding of why medium-difficulty samples are optimal.

4. **Generalizability and Scalability Experiments to Other KD frameworks on other modalities?**:
The method’s effectiveness on more diverse datasets or in non-vision tasks remains unexplored. The authors should test out the method on a range of datasets beyond vision (e.g., NLP or time-series data) to demonstrate the versatility of the approach. This would help clarify if the medium-difficulty sample selection has domain-specific benefits or if it generalizes well.

**Questions:**

1. **Logit Reshaping Justification**: How sensitive is the logit reshaping method to variations in the teacher’s predictions? Could this technique inadvertently introduce bias in scenarios with highly imbalanced classes?
2. **Generalizability Across Domains**: What challenges do the authors anticipate in adapting the propsoed method to other domains of say NLP and time-series?
3. **Efficiency vs. Performance Trade-off**: Can you elaborate on the practical trade-offs between training time and accuracy? Are there specific scenarios where maximizing efficiency might not justify the associated performance loss?

---

> ### Author Response · Authors · 2024-11-21
> **Response to Reviewer RWDY (part 1)**
>
> We are thankful for the reviewer's time in reviewing our paper.
>
> **Q1: More justification of medium-difficulty samples.**
>
> A1: Following the reviewer's suggestion, we first divide the training set into three parts (i.e., 30\% hard samples, 30\% medium samples, and 30\% easy samples selected by the teacher). Then we compute the L2-norm of the gradient of the last layer by using different samples for training and accordingly compute the standard deviation of the gradient magnitudes. In addition, we measure the discrepancy between decision boundaries formed by the student and the teacher by using the mutual information metric as follows.
>
> |     | Hard |Medium|Easy |
> |-------------|----------|----------|----------|
> | Standard Deviation |0.018  |0.022 |0.030|
> |Mutual Information |0.046  |0.051  |0.054|
>
> As shown in the table, the standard deviation of hard samples is lower, leading to a more stable training process. However, the mutual information of hard samples is also lower, which indicates the student's decision boundary of hard samples does not well align with that of the teacher. Although the mutual information is high in the case of easy samples, the training process is unstable in terms of the standard deviation of gradient magnitudes, which may result in the overfitting of the network and affect the generalizability of the student. Overall, medium-difficulty samples obtain a better trade-off between standard deviation and mutual information. The authors are thankful for the reviewer's advice for improving the paper.
>
> In addition, as suggested by the reviewer, visualizing the feature space evolution during the distillation process is an interesting idea. We will add the related visualizations in the revised manuscript for better understanding.
>
> **Q2: Analysis of logit reshaping method in highly-imbalanced datasets and unseen datasets.**
>
> A2: We thank the reviewer for the insightful comment. First, to verify if the proposed logit reshaping method works in a highly imbalanced dataset, we sample the CIFAR-100 dataset in an imbalanced manner. To be specific, we set the sampling rates to be 100\%, 99\%, 98\%,..., 1\% for class 0,1,2,...99, respectively to create a highly imbalanced dataset. Then, we use this imbalanced dataset for the student's distillation as follows:
>
> |     | w/o reshaping |w/ reshaping |w/ improved reshaping |
> |-------------|----------|----------|----------|
> | Top-1 Test Acc (\%)|56.64  |57.46 |58.09|
>
> Experimental results demonstrate that even if the dataset is highly imbalanced, the proposed logit reshaping method still improves the student's performance in terms of the top-1 testing accuracy in the second and third columns. This further verifies the generalizability of the proposed method. In addition, there is a potential way to optimize the proposed method further. Specifically, instead of aligning the teacher's logits with the class-specific distributions, we align the teacher's logits with the average distribution information of the entire dataset (i.e., the mean of class-specific distributions). In this way, we can balance the logit variance between the minority and the majority classes. Results in the last column demonstrate the effectiveness of such a reshaping technique. The above results will be added to the revised manuscript.
>
> Regarding the generalization performance on unseen datasets, we compare the transferability of different methods in Table 11 (Appendix A.5). In this experiment, we distil the students on the ImageNet dataset and transfer their representations to perform classification on two unseen datasets. Experimental results show that the proposed method significantly outperforms the other static dataset pruning methods by using logit reshaping and preserving medium-difficulty samples.
>
> **Q3: Quantitative analysis of decision boundary dynamics.**
>
> A3: Please refer to the answer for Q1 on the comparison of mutual information.
>
> **Q4: Generalizability and scalability experiments to other KD frameworks on other modalities?**
>
> A4: It is interesting to link the proposed method with applying NLP and time-series data. As the proposed method can serve as a general dataset pruning strategy by simply computing the discrepancy between the teacher's predictions and ground-truth labels, we believe it can be easily extended to other modalities and tasks, such as speech recognition and text classification. However, due to the limited computing resources, we are inclined to distribute our resources towards conducting experiments on a specific modality to comprehensively compare the effectiveness of our method and the baseline methods by re-running these methods on different datasets and obtaining results with extensive teacher-student pairs. Despite this, applying the proposed method to other modalities is intriguing. In our future work, we will explore the potential of using the proposed method in other modalities.

---

> ### Author Response · Authors · 2024-11-21
> **Response to Reviewer RWDY (part 2)**
>
> **Q5: How sensitive is the logit reshaping method to variations in the teacher’s predictions? Could this technique inadvertently introduce bias in scenarios with highly imbalanced classes?**
>
> A5: Please refer to the answer for Q2.
>
> **Q6: What challenges do the authors anticipate in adapting the proposed method to other domains of say NLP and time-series?**
>
> A6: Since the proposed method scores the training samples according to the discrepancy between the teacher's predictions and the ground-truth labels, it may be infeasible in some tasks that are trained with self-supervised learning or unsupervised learning. In this case, we may need to obtain medium-difficulty samples in an unsupervised manner. We thank the reviewer for this relevant suggestion and will explore this line of research in our future work.
>
> **Q7: Can you elaborate on the practical trade-offs between training time and accuracy? Are there specific scenarios where maximizing efficiency might not justify the associated performance loss?**
>
> A7: The reviewer raises a reasonable concern about the practical use of dataset pruning. Here are several scenarios where training efficiency is crucial. (1) In the early stages of testing ideas during research or developing proofs of concept, quick iterations are more valuable than achieving perfect accuracy. (2) Tuning the hyper-parameters of a model is time-consuming. Therefore, it is necessary to select a representative subset from the training set to accelerate the search process. (3) Training a model on a large-scale dataset with mobile devices or edge devices is infeasible due to their limited computing and storage resources. Therefore, it is necessary to eliminate the less important samples to enable training. (4) In real-world applications, such as e-commerce recommendations and autonomous driving, they need to frequently update the models due to the change in user behaviour and evolving driving environments. Therefore, selecting a coreset for periodic retraining is economically friendly.

---

> ### Comment · Reviewer_RWDY · 2024-11-24
> **Response to the Authors**
>
> Thank you for your detailed and thoughtful response to the initial reviews. I appreciate the effort you have put into addressing the concerns and providing clear explanations and justifications for the choices made in your work. I would request you to kindly include the additional experiments and analysis, done post the initial submission, into the manuscript.

---

> > ### Author Response · Authors · 2024-11-25
> > **Response to Reviewer RWDY**
> >
> > We thank the reviewer for increasing the score and for the constructive feedback. We are actively working on an update to the manuscript and will ensure that key new results, insights, and discussions generated during the rebuttal are incorporated after the end of the rebuttal stage.

---

### Official Review · Reviewer_FjLn · 2024-11-08

**Soundness:** 3
**Presentation:** 3
**Contribution:** 3
**Rating:** 6
**Confidence:** 5

**Summary:**

This manuscript introduces a dataset pruning method to enhance the efficiency of knowledge distillation (KD) by retaining a strategically selected subset of samples and reshaping logits to better align with the teacher’s knowledge.

The authors propose that selecting medium-difficulty samples strikes a good balance and helps in preserving samples that are neither too easy nor too difficult. This in turn helps to increase the efficiency and accuracy of models.

Further, they introduce a logit reshaping technique that uses global distribution information to counteract biases that dataset pruning might introduce.

The proposed method is evaluated on two benchmark datasets CIFAR-100 and ImageNet with multiple teacher-student architectures, demonstrating consistent improvements over existing pruning techniques.

**Strengths:**

1. The paper introduces a compelling approach to dataset pruning. This method looks interesting focusing on selecting medium-difficulty samples for distillation, which strikes a balance between easy and hard examples and leads to better generalization in the student model.

2. The proposed logit reshaping method effectively mitigates the distributional shift caused by pruning. This can be a key to enhancing distillation performance on pruned datasets.

3. The technique is thoroughly evaluated across multiple benchmark datasets (CIFAR-100, ImageNet) and different teacher-student architectures.

4. The proposed pruning method significantly reduces training time without sacrificing performance. This can give us with some really good solutions for real world use and even we can use this method for time consuming computationally large datasets.

5. The proposed method outperforms existing static pruning techniques (e.g., Herding, Forgetting, MoDS) and achieves comparable or superior performance to dynamic methods like InfoBatch.

**Weaknesses:**

1. The article does not provide a detailed analysis of how sensitive the method is to hyperparameter tuning. The authors should try out with different keep ratios or distillation losses.

2. This article compares with static pruning techniques but a more in-depth comparison with other knowledge distillation methods, especially self-knowledge distillation, would strengthen its position and give us a more in-depth analysis.

3. The approach is evaluated on CIFAR-100 and ImageNet, but it remains unclear how well it would scale with more complex models like ResNet family and Wide ResNets (WRN).

**Questions:**

I have some queries regarding the article----

1. How does the proposed logit reshaping method handle potential conflicts between preserving class-specific distributions and mitigating the variance of logits in a highly imbalanced dataset? Could the method be further optimized for such cases? Can I get some explanations for this?

2. In the context of medium-difficulty sample selection, how does the method ensure that the pruned dataset retains enough diversity to avoid overfitting, especially when dealing with high-dimensional feature spaces in complex models?

3. Given that the pruned dataset relies on teacher feedback to rank sample difficulty, could this process introduce potential biases in class relationships, and how might this be addressed if the teacher's knowledge is imperfect or misaligned with the pruned subset?

4. Can the authors provide some tSNE figures for better understandings.

---

> ### Author Response · Authors · 2024-11-21
> **Response to Reviewer FjLn (part 1)**
>
> We are thankful for the reviewer's time in reviewing our paper.
>
> **Q1: The article does not provide a detailed analysis of how sensitive the method is to hyperparameter tuning. The authors should try out with different keep ratios or distillation losses.**
>
> A1: We thank reviewer for this advice. Since a series of variants of KD have been proposed in recent years, it is necessary to include these methods in the evaluation. As shown in Table 3 in our manuscript, we compare the effectiveness of different dataset pruning strategies by using an advanced feature distillation loss. In addition, in Table 10 (Appendix A.4), we replace the vanilla KD loss with the recently proposed logit distillation loss (i.e., DKD) on ImageNet as follows.
>
> |Ratio | KD|DKD|
> |--|--|--|
> | r=50\%| 69.97 |**70.22**|
> | r=70\% | 70.93 |**71.61**|
>
> The results show consistent improvements under different keep ratios, demonstrating that the proposed method is compatible with the advanced distillation loss.
>
> **Q2: This article compares with static pruning techniques but a more in-depth comparison with other knowledge distillation methods, especially self-knowledge distillation, would strengthen its position and give us a more in-depth analysis.**
>
> A2: We kindly emphasize that we did compare the proposed method with several self-knowledge distillation methods in Table 5 in the manuscript. We cite the results from the paper as follows.
>
> |  | Baseline|Zipf’s LS|USKD|Ours|
> |--|--|--|--|--|
> | Top-1 Test Acc (\%)| 69.57|69.59|70.38|**70.92** |
> | Training time (h) | 39.86 |>39.86|>39.86|**36.98**|
>
> As shown in the table, even if our method uses fewer training samples and training times, it still outperforms two recently proposed self-knowledge distillation methods in terms of top-1 testing accuracy.
>
> **Q3: The approach is evaluated on CIFAR-100 and ImageNet, but it remains unclear how well it would scale with more complex models like ResNet family and Wide ResNets (WRN).**
>
> We kindly point out that some variants of ResNet (e.g., WRN and ResNet8x4) have been used as teacher networks or student networks in Tables 6-9 (Appendix A.3). Experimental results demonstrate the superior performance of the proposed method with complex models.
>
> **Q4: How does the proposed logit reshaping method handle potential conflicts between preserving class-specific distributions and mitigating the variance of logits in a highly imbalanced dataset? Could the method be further optimized for such cases? Can I get some explanations for this?**
>
> A4: We thank the reviewer for the insightful comment. First, to verify if the proposed logit reshaping method works in a highly imbalanced dataset, we sample the CIFAR-100 dataset in an imbalanced manner. To be specific, we set the sampling rates to be 100\%, 99\%, 98\%,..., 1\% for class 0,1,2,...99, respectively to create a highly imbalanced dataset. Then, we use this imbalanced dataset for the student's distillation as follows:
>
> | | w/o reshaping | w/ reshaping | w/ improved reshaping|
> |--|--|--|--|
> |Top-1 Test Acc (\%)| 56.64 |57.46|58.09|
>
> Experimental results demonstrate that even if the dataset is highly imbalanced, the proposed logit reshaping method still improves the student's performance in terms of the top-1 testing accuracy in the second and third columns. This further verifies the generalizability of the proposed method. In addition, there is a potential way to optimize the proposed method further. Specifically, instead of aligning the teacher's logits with the class-specific distributions, we align the teacher's logits with the average distribution information of the entire dataset (i.e., the mean of class-specific distributions). In this way, we can balance the logit variance between the minority and the majority classes. Results in the last column demonstrate the effectiveness of such a reshaping technique. The above results will be added to the revised manuscript.

---

> ### Author Response · Authors · 2024-11-21
> **Response to Reviewer FjLn (part 2)**
>
> **Q5: In the context of medium-difficulty sample selection, how does the method ensure that the pruned dataset retains enough diversity to avoid overfitting, especially when dealing with high-dimensional feature spaces in complex models?**
>
> A5: The reviewer raises an interesting question about the trade-off between overfitting and underfitting during the distillation process. In the following table, we divide the training set into three parts (i.e., 30\% hard samples, 30\% medium samples, and 30\% easy samples) and compare the overall training accuracies that the distilled student MobileNetV2 obtained:
>
> | |Hard|Medium|Easy|All|
> |--|--|--|--|--|
> |Top-1 Train Acc (\%)| 70.57|85.21|94.42|83.69|
>
> The above table demonstrates that the distilled student tends to under-fit the hard samples in terms of the extremely low training accuracy. On the other hand, the accuracy of the student for easy samples is much higher than that of the student for classifying the entire dataset, which may lead to the overfitting problem. The medium-difficulty samples obtain a trade-off between overfitting and underfitting. Such patterns can be observed in most of the teacher-student pairs. We conjecture that this is because the teacher's knowledge is complex, and the medium-difficulty samples are diverse enough for the student's training.
>
> **Q6: Given that the pruned dataset relies on teacher feedback to rank sample difficulty, could this process introduce potential biases in class relationships, and how might this be addressed if the teacher's knowledge is imperfect or misaligned with the pruned subset?**
>
> A6: We agree with the reviewer's opinion regarding the teacher's selection which may result in bias in the relationship between different classes. As shown in Figure 4 in our manuscript, the teacher's predictions for the preserved samples may be over-confident or under-confident in the between-class relationship according to the logit distribution. Therefore, we propose the reshaping method to mitigate the bias by using the class-wise average information to reshape the logit distribution of the preserved samples.

---

> > ### Comment · Reviewer_FjLn · 2024-11-23
> >
> > Dear Authors,
> >
> > I want a clarification. When you are showing the validation accuracies of 3 different sample types. Can you also report the training accuracies especially for the hard and medium hard samples.
> >
> > That would provide some more insights about the trade-off.

---

> > > ### Author Response · Authors · 2024-11-24
> > > **Response to Reviewer FjLn**
> > >
> > > We thank the reviewer for increasing the score. We would like to clarify that the table in Q5 shows the top-1 **training** accuracies of different types of samples. In addition, similar to the results in Table 1 in our manuscript, by using the medium-difficulty samples for distillation, the student obtains the best top-1 **testing** accuracy as follows:
> > >
> > > |        | Hard       | Medium       |Easy       |
> > > |-----------|-----------|-----------|-----------|
> > > | Top-1 Test Acc (\%)     | 46.62     | **51.71**     |51.54    |
> > >
> > > Combining the table in Q5 and the above table, we can observe that the student using medium-difficulty samples for distillation obtains a better trade-off between underfitting and overfitting on the training set, resulting in better generalization performance on the testing set. Thanks for the prompt reply. Let us know if the reviewer has any further questions.

---

### Author Response · Authors · 2024-11-28
**Revised Manuscript**

Following the reviewers' insightful comments and valuable suggestions, we have submitted a new version of the manuscript with more analysis and experiments. The newly added contents are highlighted in blue text. In summary, we incorporate the following modifications:

**We thank the Reviewer FjLn for the support and the valuable suggestions about conducting experiments on highly imbalanced datasets to complement the evaluation.**

* Results on a highly imbalanced dataset are included in Appendix A.6, demonstrating the generalizability of the proposed logit reshaping method.

* Analysis of the trade-off between underfitting and overfitting is provided in Section 3.2.

**We thank the Reviewer RWDY for the recognition of our work and the insightful comments about using mutual information and gradient variation to evaluate the proposed method.**

* More justification for using medium-difficulty samples for distillation from the perspective of gradient variation and mutual information is provided in Section 4.2.

* More visualizations on the evolution of feature space during the training process are provided in Appendix A.5.

* Results on a highly imbalanced dataset are included in Appendix A.6, demonstrating the generalizability of the proposed logit reshaping method.

**We thank the Reviewer s3Pu for recommending the relevant paper.**

* We introduce the recommended paper in Section 3 and discuss the difference with our method. In addition, we further clarify the focus of this paper at the beginning of Section 3.

* The technical details of different logit reshaping methods are described in Appendix A.3.

**We thank the Reviewer EGEk for their support and valuable suggestions to improve the readability of the manuscript.**

* Following the suggestion of the reviewer, we update Figure 2 (Figure 3 in the revised manuscript) by comparing the training accuracies of different types of samples.

* The motivation for using $M_{DIF}$ is elaborated in Section 3.1.

---

### Meta-Review · Area_Chair_zqNp · 2024-12-20

**Metareview:**

The submission proposes a dataset pruning strategy for knowledge distillation (KD), focusing on the selection of medium-difficulty samples and introducing a logit reshaping method to mitigate distributional shifts caused by pruning. The authors present a clear motivation for focusing on medium-difficulty samples. The authors conducted experiments on CIFAR-100 and ImageNet, demonstrating the method’s potential benefits.

The authors have provided detailed responses to all reviewer comments, including additional experiments, updated visualizations, and clarifications in the revised manuscript:

-Concerns about imbalanced datasets, logit reshaping sensitivity, and scalability have been thoroughly addressed, with results incorporated into the manuscript.

-Reviewer FjLn raised their score after satisfactory clarifications. Reviewer RWDY acknowledged improvements and requested additional inclusions in the revised manuscript.

-Reviewer s3Pu's Concerns: While the authors made significant efforts to address s3Pu’s concerns, the reviewer maintained their position, citing fundamental disagreements about the trade-offs inherent in dataset pruning. However, the area chair thinks it is okay for a machine-learning paper to give more attention to its methodological and theoretical parts.

**Additional Comments On Reviewer Discussion:**

Here are some suggestions for improvement:

-Further theoretical analysis could strengthen the manuscript's impact, particularly around medium-difficulty sample selection and logit reshaping.

-Expanding experiments to other modalities (e.g., NLP, time-series) in future work would demonstrate broader applicability.

---

### Decision · Program_Chairs · 2025-01-22

Accept (Poster)